



# Multi-source global wetland mapping: combining surface water imagery and groundwater constraints

Ardalan Tootchi[1], Anne Jost[1], Agnès Ducharne[1]

[1] Sorbonne Université, CNRS, EPHE, Milieux environnementaux, transferts et interaction dans les hydrosystèmes et les sols, Metis, F-75005 Paris, France

*Correspondence to*: Ardalan Tootchi (ardalan.tootchifatidehi@upmc.fr)

**Abstract.** Wetlands are important players in the Earth climate system because of their effect on ecosystems, river discharge, water quality, and through their feedback effects on atmosphere by increasing methane emission and evapotranspiration. Many datasets have been developed for open water and wetland mapping, based on three main methods: (i) compiling national/regional wetland maps; (ii) identifying inundated areas by satellite imagery; (iii) delineating wetlands as areas with shallow water table depths. There is a massive disagreement, however, between the resulting wetland extent estimates (from 3 to 21% of the land surface area). To reconcile these differences, we propose composite wetland (CW) maps consisting of two classes of wetlands: (1) regularly flooded wetlands (RFWs) which are obtained by overlapping selected open-water and inundation datasets, and cover 9.7% of the land surface area; (2) scattered groundwater wetlands (SGWs), derived either from direct groundwater modelling or simplified modelling based on the topographic index (TI), using several variants. In this framework, wetlands are defined as zones that are either inundated or where the groundwater is sufficiently close to the surface to maintain near saturated soil surface. By combining RFW and different SGW maps, seven CW maps are generated, which correspond to contemporary potential wetlands, *i.e.* the areas that would turn into actual wetlands under the present climate assuming no human influence. They are produced at the 15 arc-sec resolution (almost 500 m at the Equator) using geographic information system (GIS) tools. Two CW maps, showing the best overall match with the available evaluation datasets, are eventually selected. Wetlands in these maps respectively cover 21.1 and 21.6 % of the global land area, which is in the high end of the literature range, along with recent estimates also recognizing the contribution of groundwater-driven wetlands. The two proposed composite maps agree massively about six major wetland hotspots, which include 75 % of the global wetlands, and concentrate in the boreal, tropical, and coastal zones. The high wetland density in the tropics is brought by the SGWs, which allows detecting wetlands under dense canopy and cloud cover. Another major feature of the two CW maps, brought by the SGWs and the high resolution of the maps, is the identification of many small and scattered wetlands, which cover less than 5% of the land area, but are very important for hydrological and ecological functioning in temperate to arid areas. By distinguishing the RFWs and SGWs globally based on uniform principles, we eventually propose a simple wetland classification focused on hydrologic functioning, believed to be very useful for large-scale land surface modelling.



## 1 Introduction

Wetlands are among the most productive ecosystems in the world, comparable to rain forests and coral reefs. Natural wetlands and rice paddies are important sources of methane and carbon dioxide, with a key role in the carbon cycle (Matthews and Fung, 1987; Richey et al., 2002; Repo et al., 2007; Ringeval et al., 2012). Large wetland densities translate into lower and delayed runoff peaks, higher base flows, and increased latent heat fluxes, which directly influence climate (Bierkens and van den Hurk, 2007; Heng et al., 2016). They also serve to purify pollutions from natural and human sources, maintaining clean and sustainable water to ecosystems (Billen and Garnier, 1999; Dhote and Dixit, 2009; Curie et al., 2011; Passy et al., 2012). But despite their widely recognized importance, there is no consensus among scientists on wetland definition and their respective areal extent. Based on tens of definitions in literature, they range from areas with permanent inundation like lakes (lacustrine wetlands) with depths of several meters, to regions with relatively shallow water table (National Research Council, 1995; Kutcher, 2008; Ramsar, 2009). The first reason is that wetlands are transitional features with smooth changes in soil/water characteristics, and exact distinction between wet and non-wet lands is not easily reached. Secondly, wetlands are viewed as either aquatic, terrestrial or transitional biomes in different disciplines and therefore several water depths and wetness criteria have been suggested for their definition. Thirdly, the temporally varying nature of wetlands and human influences further complicate their definition and detection (Mialon et al., 2005; Papa et al., 2010; Ringeval et al., 2011; Sterling et al., 2013; Hu et al., 2017; Mizuochi et al., 2017). Differences between definitions have led to contrasting wetland extents and distribution among the reviewed literature in Table 1. This table shows a summary of surface water, wetland and land cover datasets or estimates with their corresponding resolution and wet fraction (wet fractions and percentages are those indicated in each study's accompanying paper or data description). In addition to disagreement in wetland definition, differences in wetland mapping methods add up to inconsistencies among wetland datasets.

The first global wetland maps were developed mainly based on compilation of regional archives and estimates. Matthews and Fung (1987) developed a 1° resolution wetland map based on vegetation, soil properties and inundation fractions, covering ~4% of the land. Finlayson et al. (1999) based their estimates on surveys, workshops and the Ramsar global review of wetland resources inventory, where wetlands covers 9.7% of the land area. Later, the global lakes and wetlands database (GLWD) was developed at 30 arc-sec resolution (~1 km at Equator) by compiling several national and regional wetland maps, leading to total wetlands covering 6.9% of land area excluding Antarctica and glaciated lands (Lehner and Döll 2004). With the widespread accessibility of satellite images in 2000s, using them for mapping water-related features has become popular, particularly since satellite imagery permits homogeneous observation of land characteristics. Previous studies using satellite imagery at visible wavelength report that 1.6 to 2.3% of Earth's land is permanently under water (Verpoorter et al., 2014; Feng et al., 2015; Yamazaki, et al., 2015; Pekel et al., 2016). Despite the high resolution of optical satellite imagery, there are large disagreements among developed wetland maps (Nakaegawa, 2012) and furthermore wetlands under densely vegetated and clouded areas are not detected (Lang and McCarty 2009). Longer wavelengths in the microwave band (*e.g.* L-band) can penetrate through the cloud and vegetation layer, providing images at 25-50 km resolution (Li and Chen, 2005; Prigent et al., 2007; Parrens et al., 2017). Fluet-Chouinard et al. (2015) developed GIEMS-D15 by downscaling the 0.25° resolution multi-satellite wetland observations of Prigent et al. (2007) using topography. However, in such downscaled datasets, wetland types where hydrology is decoupled from topography are not often well represented (*e.g.* permafrost-affected wetlands).

Wetlands derived through satellite imagery at all wavelengths almost always represent inundated areas, overlooking other types of wetlands where soil moisture is high but the top surface is not inundated. In latter, the water table depth (WTD) is close to the surface, providing direct connection between groundwater (GW) and atmosphere. One of the methods to





delineate these groundwater-related wetlands is through using WTD modelling; either direct modelling (*e.g.* Miguez-Macho and Fan 2012), or simplified modelling based on the Topographic Index (TI) of TOPMODEL (Beven and Kirkby 1979). With the advent of the global scale data set required as input to these methods such as high-resolution DEMs, they have become popular for wetland delineation. Direct GW modelling at different spatial scales have been used to map wetlands and to study

the interactions between GW and heat fluxes (Kollet and Maxwell 2008; Fan and Miguez-Macho 2011). 2D groundwater modelling requires in depth knowledge on the physics of water movement, topography at a sufficiently high resolution, climate variables, subsurface characteristics and observational constraints (Fan et al., 2013; de Graaf et al., 2015). Simplified GW models based on TI require less extensive input, and have also been widely used to map wetlands, particularly to be used as constraints for carbon fluxes modelling. Having the topography, TI can be calculated as:

$$TI = ln\left(\frac{a}{tan(\beta)}\right),$$    (1)

where $a$ is the drainage area per unit contour length and $tan(\beta)$ is the local slope at the desired pixel. TI is high over flat regions with large drainage areas, thus, high index values are often indicative of a high propensity for saturation. Other environmental characteristics like climate and soil or underground properties can also be used in TI formulation to detect

wetlands in areas where topography is not the primary driver of the water budget, like wetlands in uplands, over clayey soils or thin active layers in the permafrost region (*e.g.* Saulnier et al., 1997; Mérot et al., 2003; Hu et al., 2017).

A major challenge to identify wetlands, based on either simplified or direct WTD modelling, is to define thresholds, on TI or WTD respectively, so that all pixels having a higher TI or smaller WTD than these thresholds are accounted as wetlands. Calibrating a TI threshold for wetlands at the global scale has been shown ineffective, since it is not possible to

uniquely link TI values to soil saturation levels across different landforms and climates (Marthews et al., 2015). Consequently, TI thresholds are often calibrated to reproduce the pattern of documented wetlands in a certain region and then extrapolated for larger domains. More recently, Hu et al. (2017) produced a global wetland dataset by calibrating independent TI thresholds in all the large river basins of the world. Direct WTD modelling, at continental/global scale, is quite demanding in terms of data and CPU requirements, and has rarely been used for wetland delineation. To our knowledge, the only example is the work

of Fan and Miguez-Macho (2011) and Fan et al. (2013), who applied uniform WTD thresholds, first over North America, then globally. They propose WTD thresholds between 0 to 25 cm for wetlands and inundated areas. The resulting wetland patterns are found to be very similar for different thresholds within this range. It must be emphasized that adjusting wetland thresholds, both for directly modelled WTD and TI, always implies subjective choices.

Based on the above analysis, we propose to merge properties of multiple datasets to develop higher quality composite

maps (as in Fritz and See, 2005; Jung et al., 2006; Schepaschenko et al., 2011; Pérez-Hoyos et al., 2012; Tuanmu and Jetz, 2014). This is done for two main reasons. The first one is to reconcile disagreements and discrepancies among wetland datasets (Table 1). The second motive is to include GW related wetlands which are either covered under dense vegetation or are rarely inundated and therefore, not usually included in wetland datasets. To include all types of wetlands, we define wetlands as areas which are permanently inundated, or where the soil surface is persistently saturated or near-saturated and one can assume the

water table to be shallow. We develop composite wetland maps at 15 arc-sec resolution (within the resolution range of wetland-related datasets) by classifying them into (1) Regularly Flooded Wetlands (RFWs) and (2) Scattered Groundwater Wetlands (SGWs) representing         groundwater-driven wetlands that might not be inundated or permanent. These two classes are complementary to each other and none of them is uniquely representative for wetlands. Yet, they intersect with each other, particularly where GW discharge results in surface inundation and becomes detectable through satellite imagery. In this

framework, after explaining methods and data in Sect. 2, we first focus on the RFWs in Sect. 3. The construction of the SGW maps is detailed in Sect. 4, including the way to choose the required thresholds. The resulting composite wetland maps are then presented and compared against independent evaluation wetland datasets at global and regional scales in Sect. 5. Finally,



we conclude in Sect. 6 by discussing advantages and limitations of the proposed composite maps and giving perspective on future developments.

**2 Methods and data**

**2.1 General strategy**

We propose several composite wetland maps. Each of them is constructed using ArcGIS tools, and consists of the overlap (union) of:

- one RFW map, itself developed by overlapping three surface water and inundation datasets derived from satellite imagery;

- one SGW map out of seven, all derived from GW modelling, either direct, or simplified based on several versions of TI. In this process, many layers were developed which are summarized in Table 2 and detailed in Sect. 3 and 4. Input datasets are

presented in Sect. 2.3 and 2.4, and two independent evaluation datasets are presented in 2.5. It must be noted that lakes are excluded from our wetland maps (RFW, SGW, CW). This is because lakes are relatively deep water bodies with rather static state, both in extent and hydrological properties and as a result they do not provide an apt environment to maintain wetland specific organisms. The lakes mask used in this paper is introduced in Sect. 2.2, and the wetland percentages of the land surface area will always exclude lakes (Fig. 1a), the Caspian Sea, Greenland ice-sheet and Antarctica (unless otherwise mentioned).

For this reason, these percentages and areas might be different from those in Table 1, which are those indicated in each original paper or data description.

**2.2 Lakes**

There are several recent lakes and permanent water bodies datasets in the literature, which are not significantly different in global lakes area and spatial distribution. Here we used the HydroLAKES database (Messager et al., 2016), which was

developed by compiling national, regional and global datasets. It consists of more than 1.4 million individual lake polygons covering almost 1.7% of the total land surface area, and also contains estimations of lake volumes at the global scale for lakes with a surface area of at least 10 ha. Like in all lake databases, most of the lakes are located in the northern boreal zones (more than 60% of lakes area is north 50°N), mostly due to favourable climatic and lithologic properties (Fig. 1a).

**2.3 Input to RFW map: inundation datasets**

**2.3.1 ESA-CCI land cover**

This dataset succeeds the GlobCover dataset based on the data from MERIS sensor (on board of ENVISAT) at high resolution, along with SPOT-VEGETATION time series (Herold et al., 2015). Global land cover maps at approximately 300 m (10 arc-sec) resolution deliver data for three 5-year periods (1998-2002, 2003-2007 and 2008-2012). Extent of water bodies have slightly changed between the first 5 year period to the third one (like shrinking of Aral lake by more than 55% of the area), but

the extent of wetland classes (permanent wetlands and flooded vegetation classes) did not significantly change (the variation in wetland classes throughout these periods is less than 3% of total wetlands area). We acquired the last epoch data to represent the present state of wetlands (Fig. 1b). In this land cover dataset, legend entries that could be considered as wetlands are mixed classes of flooded areas with tree covers, shrubs or herbaceous covers plus inland water bodies. The flooded classes cover 3% of the Earth land surface.

**2.3.2 GIEMS-D15 (Fluet-Chouinard et al., 2015)**

Prigent et al. (2007) used multi-sensor satellite data including passive and active microwave measurements, along with visible and near-infrared reflectance, to calculate monthly mean inundated fractions of rather coarse equal-area grid cells (0.25° cells)



for a 12-year period between 1993 and 2004. This dataset (GIEMS) gives the minimum and maximum extent of inundated area (including wetlands, rivers, small lakes, and irrigated rice). To downscale it, Fluet-Chouinard et al. (2015) used a 15 arc-sec resolution DEM (HydroSHEDS: Lehner et al., 2008). To train a downscaling model for surface water, they used GLC2000, which is a land cover map (Bartholomé and Belward 2005) mainly obtained from optical remote sensing data (VEGETATION sensor on board SPOT-4). They developed three datasets for mean annual minimum, mean annual maximum and long-term maximum extent of inundated areas at 15 arc-sec resolution. In this study we assumed that the mean annual maximum extent was the best representative for wetlands (7.7% of the land surface area) since it stands for areas that are annually flooded at least once, while the mean annual minimum (3.9% of the land surface area) consists of river channels and permanent water bodies, and the long term maximum (10.3% of the land surface area) includes extreme events as well. In the following, GIEMS-D15 always indicates the mean annual maximum of GIEMS-D15 (Fig. 1c). A higher resolution (3 arc-sec) downscaling of GIEMS has recently been developed (Aires et al., 2017), but we overlooked as we focused our study on the 15 arc-sec resolution.

### 2.3.3 JRC surface water (Pekel et al., 2016)

The JRC surface water products are a set of high resolution maps (1 arc-sec ~ 30 m) for permanent water but also for seasonal and ephemeral water bodies. They are based on analyzing Landsat satellite images (Wulder et al., 2016) over a period of 32 years (1984-2015). Landsat satellites orbit the Earth with full-earth-coverage cycle of around 16 days. Each pixel was classified as open-water, land or non-valid observation. Open water is any pixel with standing water including fresh and saltwater. The study also quantifies the conversions, mostly referring to changes in state (lost or gained water extents, conversions from seasonal to permanent, etc.), from the beginning to end of the time series. For this study, we used the maximum surface water extent, which consists of all pixels which were under water at least once during whole period, covering almost 1.5% of the Earth land surface area (Fig. 1d).

### 2.4 Input to SGW maps

### 2.4.1 Water table depth estimates (Fan et al., 2013)

Fan et al. (2013) performed a global GW modelling using land surface elevation, the current sea level, and modelled recharge rates, as input data to estimate water table depth at 1 km resolution. This model assumes a steady flow and lateral water fluxes are calculated using the Darcy's Law and Dupuit-Forchheimer approximation for 2-D flow. The elevation is given by two DEMs at the 30 arc-sec resolution: HydroSHEDS (pixels south 60° N) and NASA-JPL ASTER global digital elevation map for the northern lands (north of 60° N). Recharge rates were modelled by the WaterGAP model (Döll and Fiedler, 2008), based on the observed rates of precipitation and simulated values of evapotranspiration and surface runoff for the contemporary period (1979-2007). To estimate subsurface transmissivity, hydraulic conductivity was assumed to decrease exponentially with depth from the thin soil layer (2 m) down to deeper layers. Hydraulic conductivities of the soil surface were derived from global Food and Agriculture Organization (FAO) global digital soil maps (5 arc-min resolution) and US Department of Agriculture (USDA) soil maps over the United States (30 arc-sec resolution). The exponential decay factor of hydraulic conductivity with depth is based on the local topographic slope (smaller decay leading to larger transmissivity when the slope is small). This factor is also adjusted for the permafrost region using an additional thermic factor (higher decay factor thus smaller transmissivity in permafrost areas). This modelling effort was constrained to water table observations available to authors (more than one million well observations). Almost 80% of the site observations compiled to constrain the model were located over the US and Canada in North America while observations were sparse over other continents, remarkably over Africa and Asia. The resulting dataset suggests vast areas with shallow water table, over the tropics, along the coastal zones, and particularly in boreal areas of North America and Asia.




### 2.4.2 Global TI (Marthews et al., 2015)

Marthews et al. (2015) produced a global map of TI at 15 arc-sec resolution, using the original formulation in Beven and Kirkby (1979) as in Eq (1). They used two global high resolution DEMs, *viz.* HydroSHEDS (Lehner et al., 2008) and Hydro1k (US. Geological Survey, 2000), at 15 and 30 arc-sec respectively. Hydro1k is used to fill the lack of information in

HydroSHEDS north of 60°N, which is outside the SRTM (Shuttle Radar Topography Mission) coverage used in HydroSHEDS. TI is calculated for every land pixel excluding lakes, reservoirs, glaciers and ice-covers. Since index values depend on pixel size which varies with latitude, they also applied the dimensionless topographic index correction of Ducharne (2009) to transform index values to equivalents for a 1 meter DEM.

### 2.4.3 CRU climate variables

To assess the impact of climate on wetlands, we used the CRU (Climatic Research Unit) monthly meteorological datasets. This dataset covers all land area from the beginning of the twentieth century (Harris et al., 2014). CRU climate time series are gridded to the 0.5° resolution, based on more than 4000 individual weather station records. To include a climate factor into the TI formulations, time series of some climate variables (*i.e.* precipitation and potential evapotranspiration based on the Penman-Monteith equation) are extracted for the contemporary period (1980-2016).

### 2.4.4 GLHYMPS (Gleeson et al., 2014)

GLHYMPS is a global permeability and porosity map based on high resolution lithology. The base lithology map is at an average resolution of ~2 km, and consists of 1,235,400 polygons (Hartmann and Moosdorf 2012). The permeability dataset and its derived hydraulic conductivity ($K_s$) estimates are also in vector format with an average polygon size of around 100 km$^2$. As noted by developers of GLHYMPS since "lithology maps represent the shallow subsurface (on the order of 100 m)",

permeability and hydraulic conductivity estimates are valid for the first 100 m of subsurface layer (Gleeson et al., 2011, 2014). Thus, transmissivity was estimated as the integral of this constant value of $K_s$ over the first 100 m below the Earth surface. We used this estimate in order to check if using available transmissivity datasets can improve global wetland identification. It should be noted that the hydraulic conductivity dataset has two versions: with and without the permafrost effect. To take into account the permafrost effect, Gleeson et al. (2014) used maps of the permafrost zonation index (PZI) from Gruber (2012) and

assigned a very low hydraulic conductivity ($K_s = 10^{-13}\ m/s$) for areas with $PZI > 0.99$. This value is homogenously applied everywhere in areas determined as permafrost. For our calculations, we rasterized the vector polygons of $K_s$ without the permafrost effect to the 15 arc-sec resolution.

### 2.5 Evaluation datasets

### 2.5.1 GLWD-3 (Lehner and Döll, 2004)

GLWD is a global lakes and wetlands dataset based on aggregating regional and global land cover and wetland maps. It contains three levels of information, the most inclusive one being GLWD-3, which is in raster format. This dataset is originally at 30 arc-sec resolution and contains 12 classes for lakes and wetlands (maps and details in the supplementary, Sect. S1 and Fig. S1). For large zones prone to water accumulation, but without solid information on existing wetlands, fractional wetland classes are defined (together they cover 4% of the land surface area). This is particularly the case within the Prairie Pothole

Region in North America and Tibetan plateau in Asia. These three classes stand for wetland densities that range between: 0-25%, 25-50% and 50-100%. Depending on the interpretation of fractional wetlands (by taking either the minimum, mean or maximum fraction of the ranges), wetlands in GLWD-3 cover between 5.8 to 7.2 % of the land surface area. In this paper, we take the mean fraction in these areas, leading to a total wetland extent of 6.3% of the land surface area.





### 2.5.2 Global wetland potential distribution (Hu et al., 2017)

Hu et al. (2017) proposed a potential wetland distribution using a "precipitation topographic wetness index", based on a new TI formulation where the drainage area is multiplied by the mean annual precipitation. This formulation is based on the concept of climato-topography index (Mérot et al. 2003) where the effective precipitation is introduced as the climate factor. In Hu et al. (2017) however, the authors claim that high evapotranspiration rates are often signs of wetlands existence and therefore, precipitation is a better indicator for wetland development than the effective rainfall. The new index is calculated at the 1 km resolution using GTOPO30 elevation data developed by USGS. Wetlands are categorized into "water" and "non-water wetlands". They correspond to two different TI thresholds, which are calibrated regionally. In each large river basin of the world (level-1 drainage area of Hydro1k), an adjustment model is trained on two random samples extracted from global land cover datasets: the water classes of the land cover datasets are used to train the model for the "water" threshold; the model for "non-water wetland" threshold is trained on the regularly flooded tree cover and herbaceous cover categories along with a global land cover dataset based on manual interpretation of Landsat Thematic Mapper (TM) and Enhanced TM Plus (ETM+). More details about land cover maps used in the adjustment model along with the wetland density map is brought in the supplementary.

The global coverage of the "water" and "non-water wetland" classes in Hu et al. (2017) is 22.6% of the earth land surface area (excluding lakes, Antarctica and the Greenland ice sheet), considering no loss due to human influence. This dataset gives the largest wetland extent within the accessible literature, with huge water-wetlands in South America and large non-water wetlands in Central Asia and Northern American continent. In this paper, we used the union of both "water" and "non-water wetland" classes of this dataset for further evaluations.

### 2.6 Preparation and integration

In order to homogenize datasets used in this study, all datasets were projected to WGS84 equi-rectangular coordination system. Also the datasets were resampled for facilitated fusion and comparison.

#### 2.6.1 Input resampling to 15 arc-sec

As discussed before the final resolution is targeted to be 15 arc-sec (~ 500 m) for consistency to available water datasets. To this end, all the datasets with different resolutions were resampled to 15 arc-sec using ArcMap (Esri, ArcGIS Desktop: Release 10.3.1 Redlands, CA) resample and aggregate tools (Data Management Toolbox and Spatial Analyst Toolbox). For dataset at coarse resolutions, each coarse pixel is disaggregated to 15 arc-sec ones keeping the same value. For datasets with higher resolutions than the target resolution (ESA-CCI land cover and JRC surface water), the aggregation process was based on a majority area function. It means that the value of the aggregated 15 arc-sec map was calculated as the value of feature covering the dominant fraction of the 15 arc-sec pixels.

#### 2.6.2 Results aggregation at coarser resolution

Each 15-arc-sec global raster contains more than 80,000 pixels along a circle of 360° of longitude, and wetlands can exhibit very small-scale patterns (*e.g.,* patchy or river-like). To facilitate visual inspection, we calculated wetland densities at 3 arc-min grids for most of the maps presented here. Additionally for some cross-comparisons like spatial correlation between two datasets, mean wetland densities at 0.5° resolution have been used. For zonal wetland area distributions, the area covered by wetlands in each 1° latitude band is displayed. All these aggregation processes were done using ArcGIS software (aggregate, resample and zonal statistics). The two independent evaluation datasets explained in Sect. 2.3 are also mapped at 3 arc-min resolution (Fig. S1 and S2).



## 3 Regularly flooded wetland (RFW) maps

### 3.1 Mapping by data fusion

To identify RFWs, we overlapped carefully selected datasets of surface water, land-cover and wetlands, namely the ESA-CCI land cover, GIEMS-D15 inundation surface and the maximum water extent in JRC surface water. These datasets were selected in order to contain different types of data acquisition since they each implement a certain definition of inundated areas. The idea behind the fusion approach chosen here is that the different wetlands identified by the different datasets are all valid, although none of them is exhaustive. As a result, using multiple inundation datasets fills the observational gap. Several other surface water datasets exist which have not been used here, either because they rely on similar methodologies, or because they mostly consist of lakes (Verpoorter et al., 2014; Yamazaki et al., 2015).

### 3.2 Geographic analysis

Overall, the RFW map covers 9.7% of the land surface area (12.9 million $km^2$) including river channels, deltas, coastal wetlands and flooded lake margins but excluding lakes (Fig. 1e). Areal coverage of the RFWs is by definition larger than the area of wetlands in all three selected datasets (Fig. 1b,c,d). The RFWs' global extent is also larger than global wetland estimates in GLWD-3. The contribution of the input inundation datasets to RFW map is fairly different. The shared fraction of the three composing elements is minuscule (5% of the total RFW land surface area coverage) showing the vast disagreement among them. 58% of RFWs are solely coming from GIEMS-D15, consisting mostly of South-East Asian floodplains, North-East Indian wet plains and rice paddies and wetland in the Prairie Pothole Region (in Northern US and Canada). On the other hand, ESA-CCI contributes mainly over the Ob River basin. JRC surface water adds small scale wetlands owed to its high resolution, particularly oases.

In terms of zonal distribution, 31% of the RFWs are concentrated north 50°N with most of the wetlands formed in the Prairie Pothole Region and Siberian lowlands due to a mixed effect of reduced drainage in permafrost areas and high precipitation rates. These northern wetlands include important sites consisting of peatlands and bogs. Cold and humid climate and also poorly drained soils of the boreal forest regions in Northern Canada on the Precambrian shield are the main hotspots of peat in the American continent. The same situation exists in western Siberian plains as well. The second zonal peak in RFW is between 20°N to 33°N, where the major contributors are the vast floodplains surrounding Mississippi, Brahmaputra, Ganges, Yangtze, Yellow River and Mesopotamian marshes. 30% of the world's RFWs are in tropical regions (20°N to 20°S) with a concentration mainly in the Amazon, Orinoco and Congo River floodplains and inundated parts of other types of wetlands like the Sudd swamp in South Sudan.

We also find that almost 40% of the RFWs are located within a 100 km distance to oceans and seas and can be assumed to be predominantly coastal wetlands (tidal fresh/saline water marshes and river deltas). These areas of dense RFWs in coastal areas are particularly noticeable in Indonesia and South-East Asian islands, northwestern coasts of the North America, eastern coasts of the US, salt marshes along the southern coasts of Chile in the subarctic region and also western coasts of tropical Africa. Yet, it must be acknowledged that a more rigorous differentiation between coastal wetlands and inland open-water wetlands would require in-situ observations or complementary soil and vegetation information.

## 4 Scattered groundwater wetland (SGW) maps

### 4.1 Mapping based on WTD

In lack of integrated, standardized and globally distributed WTD observations, a sound approach to locate groundwater-driven wetlands is to use available global direct GW modelling results and estimations. In this study, we used the global WTD estimations of Fan et al. (2013), at the 1 km resolution, and the resulting wetland map is denoted as SGW-WTD. We assumed





the mean annual WTD in wetlands to be less than 20 cm, following similar assumptions in literature (U.S. Army Corps of Engineers, 1987; Constance et al., 2007; Fan and Miguez-Macho, 2011). Thus, we filtered shallow WTD areas ( 20 cm) to derive the SGW-WTD. This results in a wetland area extending over 15% of the land surface, with large wetlands in northern areas and over the Amazon basin (Fig. 2a). We also performed a sensitivity analysis on the areal fraction of wetlands with different WTD thresholds (supplementary S2, Fig. S3 and S4), revealing that the variation in total wetland fraction is very weak for thresholds ranging between 0 and 40 cm (between 13.7 and 16.7 %), so a 20 cm threshold is a representative value. The wetland fraction, however, rapidly increases for deeper thresholds, showing that there is a clear distinction between shallow WTD areas (wetlands according to our definition) and the rest of the land.

### 4.2 Mapping based on various TIs

#### 4.2.1 Three index formulations

Topography is often not enough for wetland identification, as climate and subsurface characteristics also control water availability and vertical drainage. Using the original TI formulation Eq (1), high index zones may coincide with flat arid areas, or inversely, low index values may occur at wetland zones with small upstream drainage area over a shallow impervious layer. Several studies have been focused on improving the topographic index for wetland delineation by including other environmental factors or modifying the formulation of the index (Rodhe and Seibert, 1999; Mérot et al., 2003; Manfreda et al., 2011). In this study, we used three variants of TI. The global dataset of TI, developed by Marthews et al. (2015) at 15 arc-sec, is used to provide the original TI and as a base map to derive two other variants of the index.

The first variant index is the TCI (topography-climate index, inspired from Mérot et al., 2003):

$$TCI = ln\left(\frac{a \cdot P_e}{tan(\beta)}\right) = TI + ln(P_e), \tag{2}$$

where $P_e$ is the mean annual effective precipitation (in meters). The effective precipitation is first defined at the monthly time step, as the monthly precipitation $P_{my}$ (meters) that is not evaporated or transpired, using the monthly potential evapotranspiration $EP_{m,y}$ (meters) as a proxy for total evapotranspiration:

$$P_{m,y}^e = max(0, P_{m,y} - EP_{m,y}). \tag{3}$$

$P_e$ is then calculated as the sum of the 12 pluri-annual means of monthly effective precipitation. The required climatic variables are brought from the CRU monthly meteorological datasets (Sect. 2.2.3), over 1980-2016 to represent the contemporary period.

The second variant index (called TCTrI for topography-climate-transmissivity index) is constructed by combining the effect of heterogeneous transmissivity (Rodhe and Seibert 1999) with the above TCI:

$$TCTrI = ln\left(\frac{a \cdot P_e}{Tr \cdot tan(\beta)}\right) = TI + ln(P_e) - ln(Tr), \tag{4}$$

where $Tr$ (m²/s) is the transmissivity, calculated by integrating vertically a constant $K_s$ (saturated hydraulic conductivity in m/s) from GLHYMPS over the first 100 m below the Earth surface (Sect. 2.4.4).

#### 4.2.2 Two index thresholds

Like in many studies (Rodhe and Seibert 1999; Curie et al., 2007; Hu et al., 2017), we define TI-based wetlands as the pixels with TI above a certain threshold, itself defined to match a certain fraction of total land. This wet fraction is taken as the global wetland extent of SGW-WTD, equal to 15% (Sect. 4.1). Although global TI thresholding has not proven very satisfactory to match GLWD-3 or global inundation datasets (*e.g.* Marthews et al., 2015; Hu et al., 2017), we tried it to provide alternative SGW maps. In doing so, we prescribed the global wetland fraction to a chosen value, and the various TI formulations only change the geographic distribution of the corresponding wetlands. To apprehend the uncertainty related to the choice of the global wetland fraction, we tested two choices that are tightly linked to SGW-WTD map extent. In the first approach, we set



the TI threshold so that the wet pixels (with high index values) cover 15% of the land surface area. The corresponding maps are noted as SGW-TI(15%), SGW-TCI(15%) and SGW-TCTrI(15%) in Table 2, and show fairly different patterns (Fig. 2b,c,d). The second approach assumes that the total wetland extent, this time including both SGW and RFW, covers 15%. In other words, the TI thresholds are set so that the union of RFW and SGW-TI (TCI/TCTrI) datasets has the same extent as

SGW-WTD. The resulting groundwater-driven wetland maps are noted as SGW-TI(6%), SGW-TCI(6.6%), and SGW-TCTrI(6%), where the percentages between the parentheses refer to the SGWs extents when excluding overlapping RFWs. They vary for the different TI variants because the level of overlap between RFW and SGWs is different for different indices (Table 3). The patterns of these three maps are pretty similar to the ones of SGW-TI(15%), SGW-TCI(15%) and SGW-TCTrI(15%), with smaller densities and diminished extents (Fig. 2e,f,g).

**4.3 Comparison of the proposed SGW maps**

As shown in Table 2, seven SGW maps are developed consisting of SGW-WTD (Sect. 4.1) and six SGW-TIs (Sect. 4.2). Fig. 2 shows that the SGW-WTD map, contains high wetland extents over the northern latitudes (Fig. 2a), in contrast to other six SGW maps, particularly with regard to SGW-TI (Fig. 2b,e). Although wetlands and river valleys often coincide with pixels of high TI values since GW tends to accumulate in flat slope lowlands and depressions, using the original TI in SGW-TI results

in large diagnosed wetlands in well-known arid areas. For instance, SGW-TIs contain relatively high wetland densities in the Sahara and Kalahari Desert, Australian shield and Arabian Peninsula. Wetland densities of SGW-TIs in these regions are almost equal to wetland densities in wet regions of the world like West Siberian plain and Northern Canada (Fig. 2b,e). For a given threshold (15% in Fig. 3a), the distribution of wetlands derived from the simple TI is almost uniform over different latitudes, and independent from climatic characteristics that change within latitudes. Lower thresholds on TI variants (Fig.

2e,f,g and Fig. 3b), obviously results in a smaller wetland extent. There is no major change in the zonal distribution pattern when the wet fraction threshold changes from 15% to 6% (Fig. 2b,c,d and Fig. 3a,b).

Introducing a climate factor in the form of effective precipitation in SGW-TCI(6.6%) and SGW-TCI(15%) increases the value of the index in wet areas and decreases it in dry climates (Fig. 2c,f and Fig. 3a,b). Therefore, previously diagnosed wetlands with TI in dry climates disappear and transfer to regions with wet climates (like the Amazon basin and South Asia),

which is more realistic. On the other hand, the impact of including transmissivity for diagnosing scattered groundwater wetlands is almost a replica of low hydraulic conductivity distribution pattern in GLHYMPS, since transmissivity values sharply change in several orders of magnitude over regions with small permeability. In particular it leads to large patches of the diagnosed wetlands in North America and central Asia (Fig. 2d,g). Although sometimes these patches coincide with famous wetlands in some areas like the Pampas in South America (Fig. 2d,g and around 25°S in Fig. 3a), diagnosed wetlands extend

far beyond the actual wet regions into neighboring arid/semi-arid zones; *e.g.,* vast diagnosed wetlands in the western Siberian lowlands and Ob river basin extend southward toward Kazakh upland arid zones. In absence of precise and consistent subsurface characteristics information (particularly for cold northern areas), SGW-TCTrI shows low wetland densities in zones with known effect of transmissivity; like the Hudson Bay lowlands and the Prairie Pothole Region.

The magnitude of the intersection between SGW and RFW maps is a quantification criterion to evaluate the ability

of GW modelling to reproduce the documented inundated areas. For a given threshold (15% of global wet fraction), SGW-WTD overlaps with one third of RFWs globally (Table 3) which is almost as high as the intersection between SGW-TCI(15%) and RFW. The shared wetlands between SGW-WTD and RFW are mainly over Northern Canada and Western Siberian plains while the concentration of intersection points is high over South East Asian deltas and Indian lowlands for SGW-TCI(15%) (visual inspections between Fig. 1e and Fig. 2a,c). These intersection zones are further analyzed in Sect. 5 through using

quantitative evaluation criteria and both global and regional scale comparisons.



## 5 Composite wetland (CW) maps

### 5.1 Construction

CW maps are generated by overlapping RFW and different SGW maps. This process is done using ArcMap spatial analyst tools. Equi-resolution raster pixels of RFW and SGWs were aligned to coincide exactly on each other. By overlapping the RFW and each of the SGWs, seven composite wetland maps were obtained. As shown in Table 2, these maps are named with regard to their contributing SGW component; *e.g.*, the composite map containing RFW and SGW-TI(6%) is named CW-TI(6%). These composite wetlands cover between 15 to 22% of the land surface area. Each CW map contains RFW, SGW and also wetlands shared in both wetland classes (the intersection). The intersection between RFW and SGW maps is different depending on SGW extents and also the TI variants (Table 3). The intersected wetlands between SGWs and RFW is higher for TCI based maps and SGW-WTD compared to TI and TCTrI derived wetlands. The wetland extent in CWs is by definition larger than both RFW and SGWs and their spatial pattern depends on the contribution percentage of each component. As an example, in CW-TCI(15%), over most latitudes, the spatial pattern is similar to that of RFW, except over the tropical zones where SGWs are far more extensive than RFWs, shaping the general latitudinal pattern (Fig. 3c). Changing the percentage of SGWs based on different TI formulations (between 6 to 15%) does not considerably change the overall latitudinal pattern of the respective CWs, but the effect of RFW is larger with smaller SGW (Fig. 3d,e). In RFW there are large wetlands between 25°N to 35°N (union of dark and light blue colors in Fig. 3c), while in all of SGWs the wetland extents over these latitudes are smaller than in other regions (Fig. 3a,b).

### 5.2 Evaluation and selection

Developing any new wetland map faces the difficultly of evaluating its validity given the vast disagreements among available datasets and estimates and the substantial differences in wetland definition. In this paper, we face another difficulty since we combine several existing wetland-related datasets, so we cannot use them as independent evaluation datasets. We nevertheless compared the different CW maps to selected datasets and focused on the assessment of their similarities. In addition to the two evaluation datasets presented in Sect. 2.5, we also used SGW-WTD, since six of the developed CW maps are independent from it and it is completely independent from the other two evaluation datasets. Each of the seven composite wetland maps were compared to these three evaluation datasets using three criteria, namely spatial coincidence, Jaccard index and spatial Pearson correlation coefficient.

The first criterion, spatial coincidence (SC), shows the fraction of pixels identified as wet in an evaluation dataset that are also detected in the composite wetland dataset:

$$SC = \frac{Shared\ wetlands\ (intersection)}{Wetlands\ in\ evaluation\ dataset}.$$

It is calculated at 15 arc-sec resolution by intersecting CWs and evaluation datasets. SC ranges from 0-1 with the higher values showing more similarity between two datasets. This criterion is better suited to compare datasets with similar wetland extents. To consider the differences of wet fraction in the compared datasets, an evaluation criterion should also account for the size of the datasets in pairwise comparisons. This is achieved by the Jaccard index (JI), which is the fraction of shared wetlands in CW and evaluation dataset over the size of their union:

$$JI = \frac{Shared\ wetlands\ (intersection)}{Union\ of\ CW\ and\ evaluation\ dataset}.$$

This index helps to measure the performance of CWs with different wetland fractions. It ranges from 0 to 1: zero represents the case where the two datasets are disjoint, and one happens when two datasets are exactly the same. The third criterion is the Spatial Pearson Correlation coefficient, further referred as SPC. It is independent from the wet fractions in CWs and evaluation datasets, but is sensitive to spatial distribution pattern in pairwise comparisons. SPC values range from 0 to 1 with the higher





values showing more similarity. While the first two criteria were applied for comparison at the original 15 arc-sec resolution, SPC was calculated based on aggregated wetland densities at 0.5° resolution.

The evaluation results are shown in Fig. 4 in the form of radar charts for the RFW and the different CW maps, in comparison to the three evaluation datasets. For clarity, only two of the CW datasets and RFW are shown in color and the rest of CW datasets are all shown in grey (the complete results are presented in Table 4 and the supplementary: Tables S1 to 3). Overall, there was no CW dataset with systematically superior evaluation criteria. Two of these CW maps compared better to evaluation datasets, which are CW-TCI(15%) and CW-WTD. In particular, the evaluation criteria show that these two CW maps are closer to the evaluation datasets than the RFW map (Fig. 4). This confirms that the regularly flooded wetlands are not sufficiently representing all types of wetlands and that groundwater-driven wetlands are important contributors to global wetland distribution in the evaluation datasets. CW-WTD seems to be the most similar dataset to the evaluation datasets, but it should be reminded that, since the water table depth map of Fan et al. (2013) was used both as an input to CW-WTD and an evaluation dataset, the highest evaluation criteria values are always obtained when CW-WTD is compared to it.

Consistent and agreeing wetland datasets should have high spatial correlations with each other. But the spatial correlation between existing wetland datasets is low (*e.g.* the SPC between JRC surface water and GIEMS-D15 is 0.4 in Table 4). Looking at Table 4, the interesting point is that the SPC between the two outperforming CWs and existing wetland datasets is higher than SPC among these existing datasets. For instance the SPC between two of the composite wetland maps [*i.e.* CW-WTD and CW-TCI(15%)] and GIEMS-D15 is more than 0.7 and is higher than the same value between GIEMS-D15 and other wetland datasets. This means that the two selected CW maps reconcile the differences between existing wetland maps, whether they focus on RFWs (ESA-CCI, GIEMS-D15 and JRC surface water), or also encompass non-inundated wetlands (GLWD-3, SGW-WTD and Hu et al., 2017). It must also be noted that these two outperforming CW maps have many similarities: by construction they have almost the same wetland extent (*ca* 21%), and the combination with RFW reduced the differences found between the corresponding SGWs in boreal and tropical areas (Fig. 3). Eventually, we selected these two datasets (CW-WTD, CW-TCI(15%)) for further analysis of the corresponding wetland patterns.

## 5.3 Global scale analysis

With a total wetland extent above 21% of the Earth's land surface area (Fig. 5a,b), the two proposed CW maps are among the highest estimates of global wetland, considerably more extensive that GLWD-3 and pretty close to Hu et al. (2017). Although the global surface area of both maps are close to each other, CW-WTD shows more concentrated wetland regions, with larger wetland extents between 50°N – 63°N, presenting the largest extent for northern wetlands in the literature (Fig. 5c). CW-WTD is also showing a wider extent for these boreal wetlands, which extend further south than in CW-TCI(15%), as shown by the green belt in Fig. 5c between 40° and 60°N. This southward extension is actually stronger than the one of permafrost (Gruber, 2012), suggesting that the permafrost impact may be too strong in CW-WTD. In contrast, CW-TCI(15%) displays less extensive northern wetlands, but vast wetland zones over the tropics (10°N – 10°S) covering almost 9 million km$^2$, *i.e.* much larger than maximum wetland extents reported for these latitudes in literature (~5.6 million km$^2$ in Hu et al., 2017). This difference between the two selected maps (particularly over the humid tropical zones) is consistent with the fact that the TCI used to construct CW-TCI(15%) assumes that effective precipitation is entirely available for wetland formation, while it also contributes to surface runoff in the model used by Fan et al. (2013). Besides, the two selected maps are constrained to share the same SGW extent, leading to a tradeoff between boreal and tropical wetlands.

Overall, the latitudinal distributions of the CWs and most evaluation datasets are fairly similar, both in amplitude and pattern (Fig. 6). All evaluation datasets show larger wetland extents between 50°N to 60°N compared to any other latitude band. This is also true for CW-WTD, but in CW-TCI(15%) tropical wetlands outweigh globally. This is a characteristic feature of CW-TCI(15%) because of extensive TCI derived wetlands, as discussed above. It must be noted, however, that tropical





wetlands are important in all datasets, and that the two selected CW maps give the largest tropical wetland extents, in line with recent findings claiming that tropical wetlands have been underestimated in most wetland-related datasets (Collins et al., 2011; Gumbricht et al., 2017; Melton et al., 2013). Figure 6 also shows that RFW and GLWD-3 have similar extents over most latitude levels (except for northern subtropics 20°N -40°N), meaning that GLWD-3 mostly represents the flooded wetlands or

the parts of bigger wetlands which are inundated regularly. This is clearer over the tropics where RFWs (in South America, African savannas and Indian subcontinent) are mostly composed of river channels and floodplains (*e.g.*, Amazon, Orinoco, Congo, Niger and Brahmaputra) as shown in Fig. 1e (GLWD-3 wetland densities appear in Fig. S1b).

Back to the global maps of Fig. 5, we can identify six major wetland hotspots common to both composite wetland datasets. They are shown by rectangular windows in Fig. 5a,b, which cover 22% of the land, but include 75% of the global

wetlands: (1) North American cold lowlands and permafrost regions like the Hudson bay lowlands and the Prairie Pothole Region; (2) South American tropics and equatorial basins like the Amazon, Orinoco river basin and the Pantanal plus some subtropical southern lowlands like western plains of Parana River in northern Argentina; (3) Ob river basin and west Siberian plains; (4) African northern savanna belt, tropical floodplains and lowlands like the Congo river swamp forests, Niger Delta and the Sudd swamp, Chari and Logone river floodplains in Chad, Comeia national park in Angola and Zambezi river

floodplains in Zambia; (5) Wetlands and rice paddies in Northeastern Indian plains and Southeast Asian river deltas (*e.g.*, Ganges, Irrawaddy, Mekong, Yangtze); (6) Coastal wetlands, often within a 100 km distance to oceans and a low elevation (*e.g.*, the Maritime Southeast Asian mangroves, the everglades in Florida, and South Andes wetlands). These hotspots consist of zones with rich ecosystems where most of the important internationally recognized wetland sites (*e.g.* Ramsar convention sites) are located.

Using the windows depicted in Fig 5, the total wetland extent in these hotspots always exceeds the mean wetland extent, and can exceed 40%, as shown in Fig. 7 with the relative contribution of the different wetland components in these areas. Tests were made by varying the size of the windows to ensure the corresponding wetland extents and RFW/SGW partitioning are meaningful. The largest changes were found for the coastal wetlands, always restricted to pixels with an elevation not exceeding 100 m above the sea level. In both CW maps, the wetland fraction of the coastal zone increases when the band is narrowed, but this effect is particularly remarkable in CW-WTD. For instance while 43% of the 100 km coastal

band is made of CW-WTD, this fraction reaches 64% with a 20 km band. In all cases (CW-WTD, CW-TCI(15%), with different windows), RFW is the dominant class of wetlands over the North American lowlands, Southeast Asia and coastal areas while SGW is the main contributor in other hotspots (although in the Ob River basin it is true only for CW-WTD), confirming the importance of shallow GW to drive wetland development. Apart from these hotspots concentrating major

wetlands, the rest of the land area shows a wetland fraction below the global average, mostly composed of small scale wetlands, ephemeral stream ways or oasis. Located in temperate and rather arid areas, such scattered wetlands cover less than 5% of the land area (*ca* 7 million km$^2$ in both CW-TCI(15%) and CW-WTD), but they are very important for maintaining life. They are strongly linked to GW (dominant contribution by SGW) and all the more difficult to detect by satellite imagery as their size and saturation level may change rapidly, sometimes faster than the revisit period of the satellites.

**5.4 Regional scale analyses**

Regional evaluations have been focused on some important wetland sites where there are significant discrepancies or similarity between the developed and evaluation datasets. These include the cold and flat lowlands in Asia and North America (Ob river basin and Hudson Bay lowlands) and tropical rivers and swamps like the Sudd swamp, which are all located within the wetland hotspot zones discussed above.




### 5.4.1 Ob river basin

The Ob River is the third largest Russian river in western Siberia with a basin extending over ~$3\times10^6$ km$^2$. Despite a very large annual variability of inundated area (*e.g.* Mialon et al., 2005), it is certainly known as one of the largest wetland complex in the world (located in the wetland hotspot window number 3 in Fig. 5 a,b). The large wetland fraction in the Ob River basin

(Fig. 8), particularly that of RFW (located in the proximity of the river channels), explains the smooth temporal distribution of the discharge during the flooding period in parallel to the effect of gradual snow melting (Grippa et al., 2005). Yet, there are extensive disagreements among wetland-related datasets over this basin, where the areal coverage of wetlands ranges from 8 to 49% among the different datasets (Fig. 8). The lowest values correspond to the inundation datasets combined in RFW, which covers 20% of the basin area. There is a two-fold difference between the JRC surface water and ESA-CCI product, in which

the latter extends far beyond the river courses into the lowlands, with a pattern similar to GLWD-3 and RFW, and also very clear in CW-TCI(15%). As expected, the four datasets recognizing the contribution of GW to wetland formation (SGW-WTD; Hu et al., 2017; and our two composite maps) indicate consistently higher wetland fractions than RFW, between 33% and 49%. This reveals a significant uncertainty in the SGW contribution, which can be partly attributed to the way permafrost is taken into account. For instance, the differences of CW-WTD and CW-TCI(15%) with RFW show that SGW extends further

south in CW-WTD, which is likely linked to the temperature-based transmissivity adjustment factor in the GW model used by Fan et al. (2013). Finally, the intersection between SGW and RFW is a very large fraction of the composite wetlands, suggesting the dominant role of GW in forming inundated wetlands.

### 5.4.2 Hudson Bay lowlands

The Hudson Bay lowlands (HBL) is a vast, flat wetland area in the low subarctic regions of North America (located in hotspot

window number 1 in Fig. 5) dominated by extensive mud flats, forested bogs and fens, peatlands, swamps and marshes away from the shorelines, which together cover up to 90% of the region (Mitsch and Gosselink, 2000; Packalen et al., 2014). The HBL are entirely underlain by glacial silt and clay with low hydraulic conductivity (Hamilton et al., 1994). Below freezing temperature, most of the year, reduces drainage and water movement in the thin soil layer and facilitates wetland formation (Hamilton et al., 1994). As discussed in Sect. 2.2, less than two percent of the total land area is covered by lakes, but 62% of

them (number of lakes) are concentrated in the Northern American lowlands (Fig. 1a, Fig. 7a,b) as a result of post-glacial retreat, with 13% in the selected HBL window.

There, wetland extent ranges from 5 to 66% of the land surface according to the different datasets and CW maps (Fig. 9). The comparison underlines the inability of Landsat satellite imagery in capturing wetlands (JRC surface water, Fig. 9c), while ESA-CCI (Fig. 9a) successfully captures a large part of the wetlands from the evaluation datasets (probably through the

use of the SPOT-VEGETATION datasets to compensate the data gaps of MERIS). The RFWs extend over 28% of the area making it one of the highest inundated wetland concentrations globally (Fig. 7). Yet, like in the Ob basin, SGWs significantly contribute to both total and inundated wetland extents, as shown by the large overlap between the RFW and composite maps. Like in the Ob basin as well, wetlands are much larger in CW-WTD than in CW-TCI(15%). Since CW-WTD agrees well with the reference GLWD-3 dataset, we may conclude that CW-WTD is more realistic than CW-TCI(15%), which is consistent

with a more realistic GW modelling by Fan et al (2013), owing to an explicit dependence of the simulated WTD on the sea-level boundary condition and the permafrost (adjusted to reproduce the "observed wetland areas" in Northern America, Fan and Miguez-Macho, 2011). Yet, this conclusion is not easy to generalize, since GLWD-3 better matches the CW-TCI(15%) data set in the Ob basin.



### 5.4.3 Sudd swamp

This large wetland is located in eastern South Sudan, at around 300 m above mean sea level, and belongs to the hotspot window number 4 in Fig. 5. Recognized as a Ramsar site since 2006, it is the largest freshwater wetland in the Nile basin. More specifically, it is a floodplain of the White Nile, which receives water through Lake Victoria discharge in high water level periods but also from regional surface runoff (Sutcliffe et al., 2016). Seasonal variations and non-saturated wetlands increase the uncertainty of Sudd swamp extent estimations ranging from 7.2 - 48 $10^3$ km$^2$ (Mohamed et al., 2004 and references therein). Figure 10 shows the wetland distribution and extent in several datasets, which ranges from 1 to 27% of the rectangular window. As in the Ob and HBL, the ESA-CCI land cover has the largest wet fraction among satellite imagery-based datasets, and is the dominant contribution to RFW. As a result, these two datasets detect large inundated wetlands at the central parts of the window, but underestimate wetlands in the eastern part of the domain compared to the evaluation datasets (Fig. 10d,e,f). On the other hand, wetlands in Hu et al. (2017) are very patchy and show sharp changes of wetland density with what seems a period of 0.5°, which might be related to the coarse resolution of the precipitation data they used as input to their algorithm. The total wetland fraction is almost equal in CW-TCI(15%) and CW-WTD (between 20 and 25%), but with slightly different spatial patterns. The two datasets show a high wetland density in the central floodplain, in rather good agreement with GLWD-3 and regional estimates of saturated soil (compared to visuals in Mohamed et al., 2004; Mohamed and Savenije, 2014), but the surrounding groundwater wetlands are more scattered in CW-TCI(15%) over local flat valley bottoms since the effective precipitation is high over most of the region.

### 5.4.4 Other regions

There are numerous wetland sites with significant environmental or anthropogenic importance. These include vast wetlands and river deltas of South and Southeast Asia like Ganges, Brahmaputra, Irawaddy, Mekong and Red river. The proposed composite maps show the highest levels of overlap between SGWs and RFWs in these regions (Fig. 7), highlighting the significant surface-groundwater interactions over these river deltas which belong to hotspot number 5 in Fig. 5. Another interesting point is that SGW-WTD and SGW-TCI(15%) show a very similar distribution pattern over Southeast Asian deltas and Indonesia. Since the TCI does not depend on subsurface properties, an interpretation can be that groundwater wetland formation is almost completely explained by topography and climate in these areas. Thus, the abundancy of precipitation seems to be the primary driver for wetland formation in over these deltas, either directly or via water transfers by GW and flood propagation along the main streams.

As previously mentioned, a major advantage of including SGWs is that they contain many small and patchy wetlands (down to a scale of 15 arc-sec *ca* 500 m in the composite maps of Fig. 5). These small wetlands are not a major contribution to the total wetland extent compared to the much larger wetlands described above. Yet, they are very widespread over the globe at most latitudes. In semi-arid and arid areas for instance, many oasis and small depressions are revealed by the composite maps in North Africa and Arabian Peninsula, southern US and Central Asia, but we did not perform any detailed evaluation against local information. Although the density of the corresponding wetlands of arid regions is small (<2% of global arid zone area: 0.27-0.3 million km$^2$), such scattered wetlands are very important in maintaining ecosystems and animal habitats.

### 6 Conclusions and perspectives

Although well-known regional scale wetlands have been thoroughly inspected and their geographic and temporal properties have been analyzed in many areas of the globe, there is no universal agreement among datasets or estimates for the total global extent of the wetlands. Despite near global coverage of wetland maps based on satellite imagery, most of them only locate inundated areas, overlooking non-inundated groundwater-driven wetlands. To bridge this gap, we developed composite



wetland (CW) maps, including both the regularly flooded wetlands (RFWs) and the scattered groundwater-driven wetlands (SGWs). The corresponding maps were produced globally at high resolution (15 arc-sec, corresponding to a pixel size of *ca* 500 m at the Equator) then overlapped to form the CW maps, under the assumption that they are all relevant although not exhaustive. The same applies to the three maps of flooded areas derived from satellite imagery that were overlapped to

construct the RFW map. In contrast, we did not end up with a single SGW map, because of several uncertainties about this component.

Whether derived from simplified or direct WTD modelling (based on the topographic index, TI, or on the estimates from Fan et al., 2013), a major challenge is to define thresholds on TI or WTD to separate the wet and non-wet pixels. In line with the existing literature, we chose to define wetlands as areas where the mean WTD is less than 20 cm, and this WTD

threshold was translated into the TI threshold defining the same global wetland extent (15%). These choices necessarily remain subjective in absence of a consensual global wetland map, and the related uncertainty on wetland extent was shown to amount to a few percents of total land area, based on sensitivity analyses for reasonable values of the different thresholds. We also considered several classical variants of the TI to conclude that the TCI (topography-climate index), also favored by Hu et al. (2017) with a modified formula, was offering the best correspondence with the selected evaluation datasets. The original TI

did not capture the wetland density contrasts between arid and wet areas, while the inclusion of sub-surface transmissivity in TCTrI induced too sharp density contrasts, not always matching the recognized patterns of large wetlands. This calls for improved global transmissivity datasets, or new methods to provide a more continuous description of transmissivity than what is currently proposed based on discrete classes of lithology (Hartmann and Moosdorf, 2012; Gleeson et al., 2014) or soil texture (Fan et al., 2013).

As already stressed, we could not rigorously validate our composite wetland maps, by lack of homogeneous global observation of wetland distribution. We nevertheless compared the developed maps to three state-of-the-art wetland maps. This led us to select two composite maps, namely CW-WTD and CW-TCI(15%), showing the best overall match with the evaluation datasets. Wetlands in these maps respectively cover 27.5 and 29 million km2, i.e. 21.1 and 21.6% of the global land area (excluding lakes, Antarctica and the Greenland ice sheet). These total wetland fractions fall in the high end of the literature

range, along with recent estimates also recognizing the contribution of groundwater-driven wetlands (Fan et al., 2013; Hu et al., 2017). All these high estimates, including ours, also overlook the loss of wetlands induced by anthropogenic pressures, which is estimated to approach 30% of undisturbed, or potential wetlands (Sterling and Ducharne, 2008; Hu et al., 2017), mostly through urbanization and agricultural drainage. This is especially true for the SGWs, since most human influences on the environment were neglected in the input datasets (climate, topography, transmissivity, and sea level for the global WTD

modeling). In contrast, the RFW map was derived by overlapping satellite imagery for the contemporary period (past 5 to 34 years), thus showing most human-induced changes on surface water (including the way damming shifts wetlands to lakes or drylands, as reported by Pekel et al., 2016). Yet, the overlap of several inundation datasets with different historical depths was intended to minimize these disturbances. Eventually, the proposed CW maps are mostly representative of the contemporary potential wetlands, i.e. the areas that would turn into actual wetlands under the present climate assuming no (or limited) human

influence.

In this framework, an important conclusion is the marked similarity between the two proposed composite maps, despite their different assumptions for GW modelling. In particular, they massively agree about the six major wetland hotpots, which include 75% of the global wetlands, and concentrate in the boreal and tropical areas, as well as along the shoreline (coastal wetlands). The high wetland density in the tropics is brought by the SGWs, which allow detecting wetlands under

dense canopy and cloud cover. These conditions are tightly linked in the humid tropics, where wetlands have long been underrepresented (Collins et al., 2011; Melton et al., 2013; Gumbricht et al., 2017). The largest differences between the two proposed CW datasets are found in the boreal zones (including the two hotspots of the Prairie Pothole Region and East Siberian Taiga), although the RFWs are the dominant components. This uncertainty corresponds to the one of subsurface conditions



(transmissivity), and may be reduced owing to a better and higher-resolution description of the permafrost extent, active layer depth, hydraulic conductivity, or organic matter content.

Another major feature of the two composite wetland maps is the importance of small and scattered wetlands, as shown by the significant extent of wetlands outside the six hotspots (3.8 to 5.2% of the land area according to CW-WTD and CW-TCI(15%), respectively). This is yet another feature brought by the SGWs, since these small wetlands are often difficult to detect by satellite imagery techniques, especially for the non-inundated or ephemeral ones, with a size varying rapidly compared to revisit period of the satellites. The DEM resolution used here (~500 m) is fine enough to detect many of these small wetlands, but this resolution remains coarser than patchy wetlands in small depressions or riparian zones along small order streams. This may lead to overestimate the extent of these small wetlands, since the pixels are either fully wet or fully non-wet. A better delineation is expected from higher resolution DEMs, like the MERIT (Multi-Error-Removed Improved-Terrain) DEM of Yamazaki et al. (2017) offering a worldwide 3-arc resolution.

By distinguishing the RFWs and SGWs, we eventually proposed a simple wetland classification focused on their hydrologic functioning. Compared to classical wetland classifications, strongly based on floristic inventories and habitat typologies (*e.g.* Zoltai and Vitt, 1995; Finlayson et al., 1999; Lehner and Döll, 2004; Herold et al., 2015), we separated areas where wet conditions at the surface are primarily driven by flooding, or GW inputs, or both where the two classes intersect. Since the underlying principles and input datasets are valid globally, this classification is believed to be very useful for land surface hydrological modelling. In particular, we intend to use it in the ORCHIDEE land surface model (Krinner et al., 2005; Ducharne et al., 2017) to describe the areas where GW convergence from the uplands to the lowlands sustains enough moisture to enhance the local evapotranspiration and related land-atmosphere feedbacks (*e.g.* Bierkens and van den Hurk, 2007; Maxwell et al., 2007; Vergnes et al., 2014; Wang et al., 2017). Further modelling applications include methane emissions and peatland dynamics (*e.g.* Gedney et al., 2004; Ringeval et al., 2012; Qiu et al., 2018), which are tightly coupled and constitute a major uncertainty for future green-house gas emissions.

## 7 Data Availability

The raster datasets of the two proposed composite wetlands (CW-WTD and CW-TCI(%)) along with the RFW at 15 arc-sec resolution are freely available at http://www.metis.upmc.fr/en/******.

## Acknowledgment

This research is a part of the PhD project of Ardalan Tootchi, funded by "Région Ile de France" via the "Réseau francilien de recherche sur le développement soutenable".

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





**Table 1: Summary of water body, wetland and related proxy maps and datasets from the literature.**

| Name | Resolution | Type of acquisition | Wetland extent including lakes | |
|---|---|---|---|---|
| | | | (million km²) | % of the land |
| Maltby and Turner (1983) | - | Based on Russian geographical studies | 8.6 | 6.6% |
| Matthews and Fung (1987) | 1 degree | Development from soil, vegetation and inundation maps | 5.3 | 4.0% |
| Mitsch and Gosselink (2000) | Polygons | Gross estimates, Combination of estimates and maps | ~20 | ~15.3% |
| GLWD-3 Lehner and Döll (2004) | 30 arcsec ~1km | Compilation of national/international maps | 8.3 - 10.2 | 6.3 - 7.8% |
| GLC2000 Bartholomé and Belward (2005) | 1 km at Equator | SPOT vegetation mission satellite observations | 5.7 | 4.4% |
| ESA-CCI land cover Herold et al. (2015) | 10 arcsec ~300m | Multi sensor: SPOT vegetation, MERIS products | 6.1 | 4.7% |
| Fan et al. (2013) | 30 arcsec ~1km | Groundwater modelling | ~19.3 | ~17% |
| GIEMS-D15 Fluet-Chouinard et al. (2015) | 15 arcsec ~460m | Multi-sensor: SSM/I, ERS-1, AVHRR, Downscaled from a 0.25° wetland map | 6.5 – 17.3 | 5.0 - 13.2% |
| G3WBM Yamazaki et al. (2015) | 3 arcsec ~90m | Satellite imagery: Landsat | 3.7 | 2.8% |
| JRC Surface water Pekel et al. (2016) | 1 arcsec ~30m | Satellite imagery: Landsat, including maximum water extent and interannual occurrence | 2.8 – 4.4 | 2.1 - 3.4% |
| Hu et al. (2017) | 1 km | Development based on topographic wetness index and land-cover | 27.7 | 21.1% |

**Table 2: Layers of wetlands constructed in the paper, their definition and the subsection where they are explained. Total land area for wetland percentages excludes lakes, Antarctica and the Greenland ice sheet.**

| Layer | | | Definition | Wetland percentage | Explained in |
|---|---|---|---|---|---|
| **RFW** (Regularly Flooded Wetlands) | | | Union of three inundation datasets (ESA-CCI, GIEMS-D15, JRC surface water) | 9.7% | Sect. 3.2 |
| **SGW** (Scattered Groundwater Wetland) | **WTD** | | Pixels with water table depth less than 20 cm (Fan et al. 2013) | 15% | Sect. 4.1 |
| | **TI** | **(6%)** | Pixels with highest Tis, covering 15% of total land when combined with RFW | 6% | Sect. 4.2 |
| | | **(15%)** | Pixels with highest TIs values covering 15% of land | 15% | |
| | **TCI** | **(6.6%)** | Pixels with highest TCIs, covering 15% of total land when combined with RFW | 6.6% | |
| | | **(15%)** | Pixels with highest TCI values covering 15% of land | 15% | |
| | **TCTrI** | **(6%)** | Pixels with highest TCTrI, covering 15% of total land when combined with RFW | 6% | |
| | | **(15%)** | Pixels with highest TCTrI values covering 15% of land | 15% | |
| **CW** (Composite Wetland) | **WTD** | | Union of RFW and SGW-WTD | 21.1% | Sect. 5 |
| | **TI** | **(6%)** | Union of RFW and SGW-TI(6%) | 15% | |
| | | **(15%)** | Union of RFW and SGW-TI(15%) | 22.2% | |
| | **TCI** | **(6.6%)** | Union of RFW and SGW-TCI(6.6%) | 15% | |
| | | **(15%)** | Union of RFW and SGW-TCI(15%) | 21.6% | |
| | **TCTrI** | **(6%)** | Union of RFW and SGW-TCTrI(6%) | 15% | |
| | | **(15%)** | Union of RFW and SGW-TCTrI(15%) | 22.3% | |



**Table 3: Percent of overlap between SGW and RFW (percent of total pixels).**

| Groundwater-driven wetland layer | Intersecting with RFW | Non-intersecting with RFW |
|---|---|---|
| SGW-TI(6%) | 0.7% | 5.3% |
| SGW-TCI(6.6%) | 1.3% | 5.3% |
| SGW-TCTrI(6%) | 0.7% | 5.3% |
| SGW-TI(15%) | 2.5% | 12.5% |
| SGW-TCI(15%) | 3.6% | 11.4% |
| SGW-TCTrI(15%) | 2.4% | 12.6% |
| SGW-WTD(15%) | 3.8% | 11.2% |

5     **Table 4: Correlation between the developed and reference datasets (wetland fractions in 0.5° grid-cells). The highest three values in each column are shown in bold format, and grey cells give the values used in Fig. 4.**

| Dataset name | ESA-CCI | GIEMS-D15 | JRC surface water | RFW | GLWD-3 | SGW-WTD | Hu et al. (2017) |
|---|---|---|---|---|---|---|---|
| SGW-TI(15%) | -0.05 | 0.11 | -0.11 | 0.05 | 0.27 | 0.25 | 0.39 |
| SGW-TCTrI(15%) | -0.02 | -0.01 | -0.11 | 0.01 | 0.17 | 0.30 | 0.24 |
| SGW-TCI(15%) | 0.07 | 0.27 | -0.07 | 0.25 | 0.25 | **0.60** | 0.36 |
| SGW-WTD | 0.30 | 0.30 | 0.09 | 0.33 | **0.40** | **1.00** | **0.47** |
| CW-TI(6%) | 0.59 | 0.68 | 0.45 | **0.95** | 0.30 | 0.35 | 0.32 |
| CW-TCTrI(6%) | 0.55 | 0.62 | 0.43 | 0.89 | 0.32 | 0.41 | 0.36 |
| CW-TCI(6.6%) | 0.60 | 0.69 | 0.48 | **0.98** | 0.31 | 0.45 | 0.35 |
| CW-TI(15%) | 0.64 | 0.71 | 0.51 | 0.72 | 0.30 | 0.41 | 0.40 |
| CW-TCTrI(15%) | 0.58 | 0.62 | 0.48 | 0.64 | 0.30 | 0.45 | 0.37 |
| **CW-TCI(15%)** | **0.70** | **0.79** | **0.58** | 0.79 | 0.32 | 0.57 | 0.39 |
| **CW-WTD** | 0.68 | **0.72** | 0.53 | 0.75 | 0.38 | **0.72** | **0.44** |
| ESA-CCI | **1.00** | 0.39 | **0.71** | 0.62 | 0.29 | 0.30 | 0.25 |
| GIEMS-D15 | 0.39 | **1.00** | 0.40 | 0.71 | 0.26 | 0.30 | 0.27 |
| JRC surface water | **0.71** | 0.40 | **1.00** | 0.50 | 0.09 | 0.09 | 0.12 |
| RFW | 0.62 | 0.71 | 0.50 | **1.00** | 0.30 | 0.33 | 0.29 |
| GLWD-3 | 0.29 | 0.26 | 0.09 | 0.30 | **1.00** | 0.40 | 0.42 |
| Hu et al. (2017) | 0.25 | 0.27 | 0.12 | 0.29 | **0.42** | 0.47 | **1.00** |





**Figure 1: Density of lakes, regularly flooded wetlands and components of the latter (percent area in 3 arc-min grid-cells). For zonal wetland area distributions (right side charts), the area covered by wetlands in each 1° latitude band is displayed.**



**Figure 2: Density of scattered groundwater wetlands based on different approaches (percent area in 3 arc-min grid-cells). For zonal wetland area distributions (right side charts), the area covered by wetlands in each 1° latitude band is displayed.**





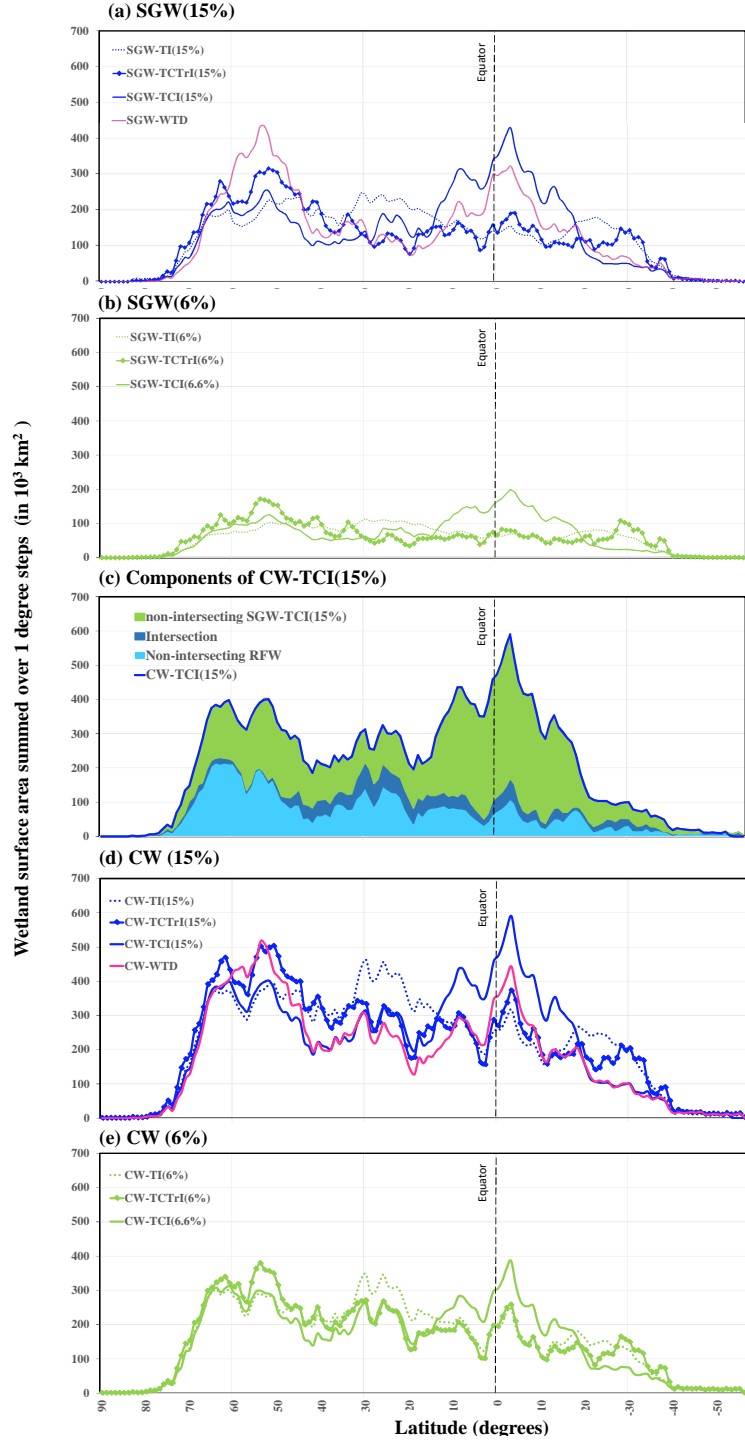

**Figure 3: Latitudinal distribution of different wetland maps; (a,b) SGWs, (c) components of CW-TCI(15%) and their intersection, (d,e) CWs. The wetland areas along the y-axis are surface areas in each 1° latitudinal band.**





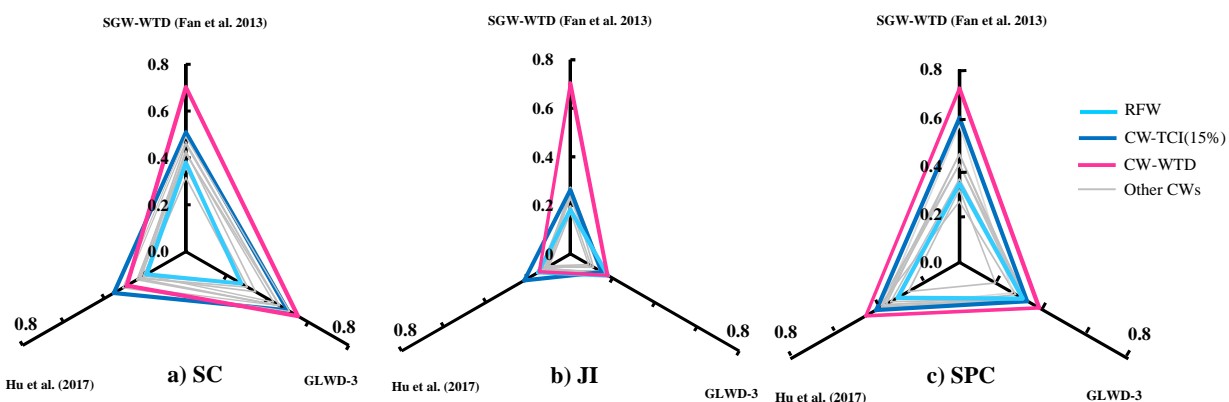

**Figure 4: Evaluation criteria between all generated composite wetland maps and evaluation datasets (SPC is calculated for wetland densities in 0.5° grid-cells)**





**Figure 5: Wetland density (as percent area in 3 arc-min grid-cells): (a) in CW-WTD, (b) in CW-TCI(15%), (c) difference between them. Numbers on (a) and (b) refer to the wetland hotspot windows explained in Sect. 5. For zonal wetland area distributions (right side charts), the area covered by wetlands in each 1° latitude band is displayed.**





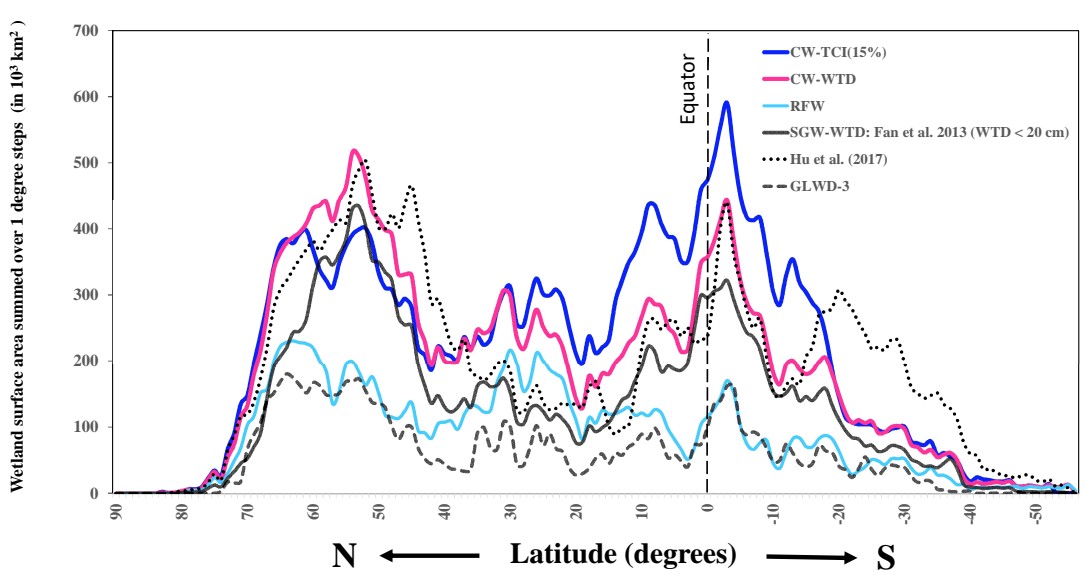

**Figure 6: Latitudinal distribution of the selected CWs and evaluation datasets. The wetland areas along the y-axis are surface areas in each 1° latitudinal band.**





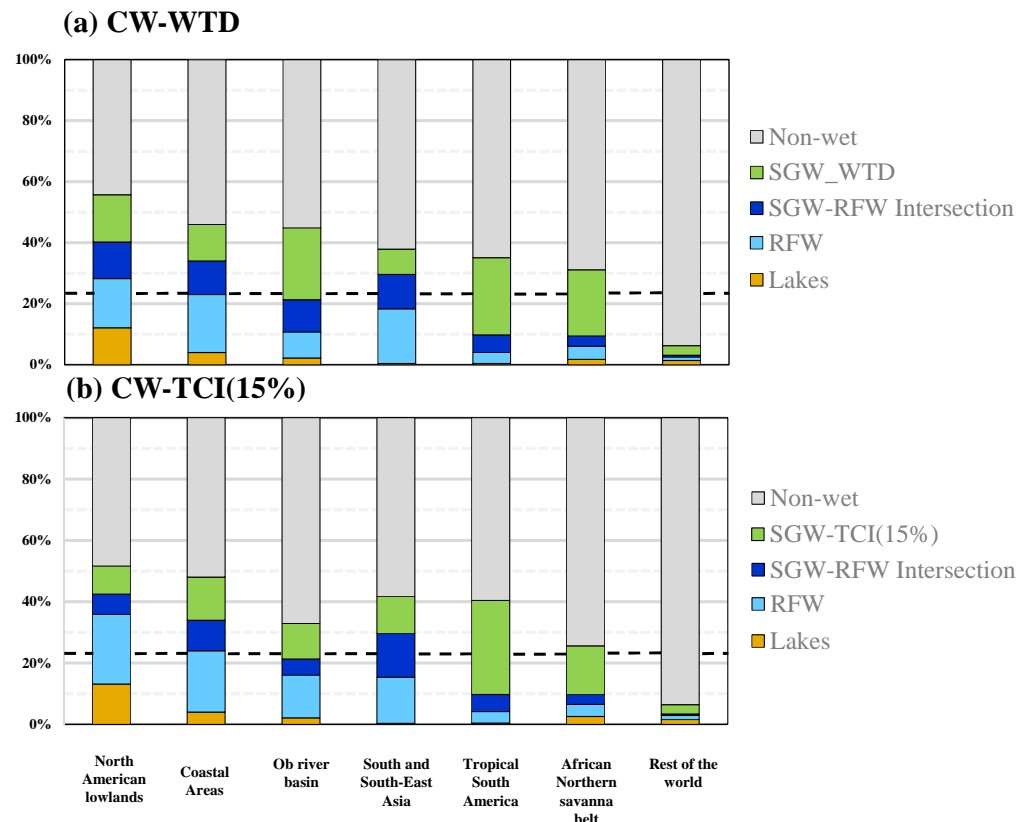

**Figure 7: Contribution of non-wet areas, lakes, RFW, SGW, and their intersection in the wetland hotspot window shown in Fig. 5: (a) in CW-WTD, (b) in CW-TCI(15%). The dashed line shows the average global wet fraction, equal to 21.1% in (a) and 21.6% in (b).**







**Figure 8: Maps of the Ob River basin wetlands according to different water and wetland datasets: (a, b, c) components of RFW, (d, e, f) evaluation datasets, (g, h, i) datasets generated in this study. The panels also give the mean areal wetland fraction of each dataset in the study area (using the mean fraction of each fractional wetland class of GLWD-3, cf. Sect. 2.5.1). The bounds of the basin are taken from the HydroBASINS layer of HydroSHEDS.**







**Figure 9: Maps of the Hudson Bay Lowlands wetlands according to different water and wetland datasets: (a, b, c) components of RFW, (d, e, f) evaluation datasets, (g, h, i) datasets generated in this study. The panels also give the mean areal wetland fraction of each dataset in the study area (using the mean fraction of each fractional wetland class of GLWD-3, cf. Sect. 2.5.1). The bounds of the study area are (48°-56°N, 76°-86°W).**





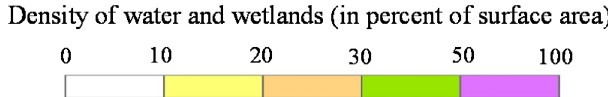

**Figure 10: Maps of the Sudd swamp wetlands according to different water and wetland datasets: (a, b, c) components of RFW, (d, e, f) evaluation datasets, (g, h, i) datasets generated in this study. The panels also give the mean areal wetland fraction of each dataset in the study area (using the mean fraction of each fractional wetland class of GLWD-3, cf. Sect. 2.5.1). The bounds of the study area are (4.5°-14°N, 24.5°-34°E).**