# Peer review of "Multi-source global wetland mapping: combining surface water imagery and groundwater constraints"

_Hydrology and Earth System Sciences, 2018_

## Referee Comment (RC1) · Anonymous Referee #1 · 9 Mar 2018

The manuscript by Tootchi et al describes a series of methodologies used to develop a global wetland dataset to be used for hydrologic and biogeochemical studies. The manuscript and methodology is presented in some detail and is thus straightforward to follow. The challenge of mapping wetland area comes down to inadequacy of a single dataset, problems with definitions, and the issue of double counting, see (Poulter et al. 2017). The authors address the first challenge by using the state of art datasets, although they miss the important contribution of SWAMPS (Schroeder et al. 2015), which should be referenced for completeness. However, I have concerns on the definition of wetland and whether their final product may continue the issue of double counting.

[Figure]

The main confusion I have is on the author's use of the JRC inland waters dataset. It appears that the authors have used the JRC inland waters to add in these features that are between 30 meters and 10 hecatares in size, and with HydroLakes being used to mask inland waters larger than 10 ha. Thus the authors final product includes rivers, ponds and small lakes. These are not vegetation wetlands and have a very different hydrologic and biogeochemical role than vegetated wetlands. The author's objective described in the Introduction is to map vegetated wetlands, and thus the inclusion of JRC inland waters was a source of confusion to me and seems inappropriate.

Thus the author's wetland area estimate is far higher than most of the existing literature except one recent paper (Hu 2017) which claims 'wetland potential' is $\sim$ 29.8 Mkm2. It is difficult to be convincing that there are > 10 Mkm2 land areas has not been detected and captured by previous studies from satellite observations and inventories unless the authors provide regional validations, especially for the regions that are not covered by previous studies.

Minor The Introduction should not cite Prigent in the discussion of L-band applications of radar – their products are C band or short wavelengths.

A regional validation for Amazonia is needed as this is the largest wetland regions in the world and there are several regional datasets can be used for validation, e.g., work of Melack, Hess and others using PALSAR

The Hu (2017) map in Figure 10 seems has some artificial barriers. Why is that? Have you considered it in your comparison or validation?

Poulter, B., P. Bousquet, J. G. Canadell, P. Ciais, A. Peregon, M. Saunois, V. K. Arora, D. Beerling, V. Brovkin, C. D. Jones, F. Joos, N. Gedney, A. Ito, T. Kleinen, C. Koven, K. McDonald, J. R. Melton, C. Peng, S. Peng, C. Prigent, R. Schroder, W. Riley, M. Saito, R. Spahni, H. Tian, L. Taylor, N. Viovy, D. Wilton, A. Wiltshire, X. Xu, B. Zhang, Z. Qiuan, and Z. Zhang. 2017. Global wetland contribution to 2000-2012 atmospheric methane growth rate dynamics. Environmental Research Letters 12:094013.

[Figure]

Schroeder, R., K. C. McDonald, B. D. Chapman, K. Jensen, E. Podest, Z. D. Tessler, T. J. Bohn, and R. Zimmermann. 2015. Development and Evaluation of a Multi-Year Fractional Surface Water Data Set Derived from Active/Passive Microwave Remote Sensing Data. Remote Sensing 7:16688-16732.

---

## Referee Comment (RC2) · Anonymous Referee #2 · 11 Mar 2018

This manuscript develops several candidate global wetland maps by combining various published maps as well as a few new Topographic (Wetness) Index based maps. It finds that the total wetland extent varies widely depending on the combination chosen.

Main comment:

I commend the authors on a great deal of GIS data processing, but I struggled to identify a hypothesis or main insight from this study. This study appears not so much intended as a scientific study but rather a mapping effort. In that case, there should be an independent validation effort to determine the accuracy of the derived product.

Evaluation of the mapped wetlands is done using the maps produced by Lehner & Döll

(2004) and Hu et al. (2017). It is not explained why it would be reasonable to put more faith into those mapping efforts than in any of the other, in other words, why they would be a suitable reference 'truth'. If all their data is of better quality, then why not just use it instead?

This study does need more robust validation using higher quality wetland mapping. One possibility is that some of the data used in the Lehner & Döll (2004) mapping are of much higher quality than the candidate data sets, but this is not discussed. If so, then validation and accuracy assessment may be possible for those selected regions where such more accurate mapping is available.

Another possibility would be to create generate a stratified randomised sample of locations in different probability classes and use very high-resolution imagery (e.g. Google Earth) to visually develop a validation data set. This type of validation effort is fairly standard for mapping studies, but it may not always be easy to identify wetlands from high-resolution imagery or even photos.

Using the results from such a validation, you might be able to assign a qualitative weighting to the candidate maps and merge them into a single global map of wetland probability.

In summary, as a data production effort, this manuscript does not help to reduce inconsistencies between existing mapping efforts. Without a thorough validation and accuracy assessment, it does not provide a demonstrable advance.

Other comments:

1) I don't understand the meaning of the word 'scattered' groundwater wetlands. P3, l38 suggests the two classes are complementary yet intersect. This cannot both be correct in the formal sense, and indeed they are not complementary. What about irregularly flooded wetlands, flooded groundwater wetlands, contiguous groundwater wetlands, scattered flooded wetlands, etc? In other words, the conceptual classification
framework needs more thought. A broader distinction between surface water- and groundwater-dominated wetlands might work better, for example.

2) The language needs more work. It is sometimes incorrect, sometimes imprecise or ambiguous. Incorrectly used or imprecise words used include: massive, detecting, pretty, players, popular, believed, replica, patches. The grammar is also lacking in places and needs checking (e.g., "in the high end", "In latter", "Back to"

3) P4,l18 – pls describe what data sets there are and how you know that they are not significantly different.

4) p4,l30 - Pls describe what method is used to delineate wetlands in the ESA-CCI product.

5) p4,l35 and further on – journal may not accept a reference in a section header, even less so if you don't repeat it in the main text.

6) p9,l34 – sound circular to me, pls explain if it isn't.

---

## Author Comment (AC1) · 17 May 2018

We thank the Referee #1 for his/her comments and suggestions. They will, for sure, help enhance the quality of the work and clarify ambiguities. We will address each block of comments in the following with the comments in black, replies in blue and modifications of the manuscript in green.

As a preamble, we would like to stress that, to develop the validation of our wetland maps recommended by both reviewers, we profoundly changed the structure of the manuscript. A summary of the major modifications to the manuscript with regard to the referee's comments are listed below:

- Introduction (stated and clarified the scientific objectives)
- New section: Sect. 2.1 Wetland definition
- Sect. 2.2 Lakes (revised to explain the lakes mask)
- Section 2.5 (added regional validation datasets)
- New section: **Section 3** (containing Sect. 3,4 and 5.1 of the previous manuscript)
- New section: **Section 4** (quantitative global and regional validation, regional scale analysis, including Sect. 5.2 and 5.4 of the previous manuscript)
- New section: **Section 5** (Discussion on the characteristics of composite wetland maps and the role of groundwater-driven wetlands)

All the figures and tables will be provided as a supplementary to this document.

**Comment 1:** "The manuscript by Tootchi et al describes a series of methodologies used to develop a global wetland dataset to be used for hydrologic and biogeochemical studies. The manuscript and methodology is presented in some detail and is thus straightforward to follow. The challenge of mapping wetland area comes down to inadequacy of a single dataset, problems with definitions, and the issue of double counting, see (Poulter et al. 2017). The authors address the first challenge by using the state of art datasets, although they miss the important contribution of SWAMPS (Schroeder et al. 2015), which should be referenced for completeness."

**Reply:** We agree with the referee. We will cite SWAMPS in the modified version as state of the art satellite imagery based wetland datasets. We tried to use high resolution datasets to produce (and evaluate) our developed wetland maps. SWAMPS dataset (Schroeder et al. 2015) was not preferred due to its rather coarse resolution (25 km).

**Modification:**

**• Introduction, end of first Paragraph:**

"Differences between definitions have led to contrasting wetland extents and distribution among the reviewed literature in Table 1, and may also result in the double counting of wetland biogeochemical emissions from deep lakes, saline and artificial wetlands (Poulter et al., 2017)."

• Introduction, end of 2nd Paragraph:

"Longer wavelengths in the microwave band (e.g. L and C band) penetrate better through the cloud and vegetation layer and provide dynamic observations of inundated zones, usually with a trade-off between high resolution with low revisit or domain extent (Li and Chen, 2005; Hess et al., 2015), and coarse resolution with high revisit up to global coverage (Prigent et al., 2007; Papa et al., 2010; Schroeder et al., 2015; Parrens et al., 2017)." • Table 1: Several rows are added including one for SWAMPS dataset.

**Comment 2:** "However, I have concerns on the definition of wetland and whether their final product may continue the issue of double counting. The main confusion I have is on the author's use of the JRC inland waters dataset. It appears that the authors have used the JRC inland waters to add in these features that are between 30 meters and 10 hectares in size, and with HydroLakes being used to mask inland waters larger than 10 ha. Thus the authors' final product includes rivers, ponds and small lakes. These are not vegetation wetlands and have a very different hydrologic and biogeochemical role than vegetated wetlands. The author's objective described in the Introduction is to map vegetated wetlands, and thus the inclusion of JRC inland waters was a source of confusion to me and seems inappropriate.

**Reply 2:** We did not exclusively seek to map vegetated wetlands. In our manuscript wetlands are zones of interaction between surface and groundwater, where the vegetation might not be present (or enough dense) due to climatic situation.

The double counting issue is a specific problem with regard to methane emission modelling which is not our primary objective. We therefore do not intend to exclude (or classify) wetlands of different ecological characteristics (like freshwater/saline wetlands, rice paddies and other anthropogenic wetlands). The final wetland maps of the study contain wetlands, indeed including small ponds and lakes, which were not detectable at the final resolution of the wetland product (15"). We advise those who wish to use the products of our study for modelling purposes (methane emission) to use a compatible and coherent lake mask (adapted to their resolution) and to use the resampling tools with "dominant fraction" methodology to avoid double counting of lakes' emissions. Future work should include further wetland classification (similar to those in GLWD) to facilitate the use of dataset for geochemical purposes.

To exclude static and deep lakes we used the HydroLAKES. Although several lake datasets exist in literature, they do not often differentiate lakes and and other waterbodies (GLOWABO: Verpoorter et al., 2014; G3WBM: Yamazaki et al., 2015). Following to similar assumptions in other maps (Ramsar convention, Fan et al., 2013, Hess et al., 2015, Hu et al., 2017) we include them as as part of wetlands

JRC surface water dataset represents smaller surface water elements than the resolution of composite wetland map. Resampling JRC surface water to 15 arcsec leads to filtering almost half of water bodies (~3% at 1 arcsec to ~1.5% at 15 arcsec). As a result, the wetland fraction which is coming solely from the JRC dataset is only 0.4% of the global land surface in our wetland products (compared to the land coverage fraction of the three maps proposed in the manuscript: 9.7%, 21.1% and 21.6%). The added value of including JRC surface water is the small wetlands and oasis concentrated enough to cover more than 50% of a 15 arcsec pixel over limited areas as: surrounding of Sebkha d'Oran lake in Algeria, wetlands downstream of Guadalquivir river in Southern Spain and some small wet plains in Southern Tashkent of Uzbekistan.

**Modifications 2:**

• Fifth Paragraph of Introduction:

"The scientific objective of the present work is to contribute to the long lasting effort for wetland delineation at the global scale since Matthews and Fung (1987). Based on the above analysis, our rationale is that inundated and groundwater-driven wetlands are both important contributions to total wetlands, and both need to be accounted for to realistically capture the wetland patterns and extents. This leads to define wetlands as areas which are persistently saturated or near-saturated, because they are regularly subject to inundation or shallow water tables."

• 2.1 Wetland definition and general mapping strategy" :

"The wetland definition behind the proposed composite maps is focused on hydrological functioning, and is not restricted to vegetated wetlands. We aim at including both seasonal and permanent wetlands, as well as the shallow surface water bodies (including rivers, both permanent and intermittent), since these two types of objects are often hydrologically connected. As a result, the transition between them is not sharp and varies seasonally, so they share many environmental properties, and they are difficult to separate based on observations (either in situ or remote). By including the shallow surface water bodies (in the RFW map), we follow the Ramsar classification, but we depart from it regarding large permanent lakes, which are excluded from the composite wetland maps (section 2.2), since they are very specific water bodies, with distinct hydrology and ecology compared to wetlands and even rivers. The groundwater-driven wetlands, in contrast, can be wet without being ever inundated owing to the presence of a shallow water table. As further discussed in section 3.2, they are defined in this study as areas where the mean annual WTD is less than 20 cm, following similar assumptions in the literature (U.S. Army Corps of Engineers, 1987; Constance et al., 2007; Fan and Miguez-Macho, 2011).

Another feature of the proposed wetland maps is that they are static. As stated in Prigent et al. (2007), they represent therefore the "climatological maximum extent of active wetlands and inundation" (for CW and RFW respectively), i.e. the areas that happen to be saturated or near saturated frequently enough to develop the specific features of wetlands (high soil moisture over a significant part of the year leading to reducing conditions in some horizons, and specific flora and fauna). Potential applications of these static maps are to assign specific hydrological properties or processes to the places identified as wetlands or floodplains. As such, the CW can be seen as the spatial support of a particular "hydrotope" (Gurtz et al., 1999; Hattermann et al., 2004), i.e. the hydrological analog of plant functional types (PFTs) for vegetation properties and processes. Other applications regard the estimation of methane production or denitrification by wetlands, especially if combined to a dynamic modelling of the saturation degree within the wetland fractions. In this particular framework, it must be underlined that the CW maps do not completely solve the so-called "double counting" issue for methane emissions (Poulter et al., 2017), since they include shallow permanent flooded areas and rivers (including the small ones, frequently intermittent), for which no dedicated exhaustive dataset is currently available (Raymond et al 2013; Schneider et al., 2017)."

• 2.2 Lakes

"To distinguish large permanent lakes from wetlands, we used the HydroLAKES database (Messager et al., 2016), which was developed by compiling national, regional and global datasets. It consists of more than 1.4 million individual polygons for lakes with a surface area of at least 10 ha, covering 1.8% of the land surface area. This value is smaller than in other recent databases accounting for smaller water bodies: 2.5% in G3WBM (Yamazaki et al., 2015) for water bodies above 0.8 ha; 3.5% in GLOWABO (Verpoorter et al., 2014) for those above 0.2 ha. These two datasets do not differentiate lakes, and using them as a lake mask would lead to exclude areas that can be considered as wetlands according to Sect 2.1, like shallow water bodies in important wetland sites, like in the Ob river basin, Indonesian mangroves, or Ganges floodplains. These areas can be recognized as wetlands by the datasets underlying the RFW map (Sect. 2.3), so we based the lake mask on HydroLAKES to conserve these wetlands. It must also be noted that the small water bodies tend to be overlooked after dominant resampling to the 15 arc-sec (Sect 2.6), unless they are sufficiently numerous in a pixel. This explains why the lake mask shown in Fig. 1a covers only 1.7% of the land area, compared to 1.8% in the original HydroLAKES database. This map also shows that most of the lakes are located in the northern boreal zones (more than 60% of lakes area is north 50°N), in agreement with the other lake databases."

• First Paragraph of Geographic Analysis (RFW) :

*"JRC surface water adds small scale wetlands owed to its high resolution, particularly oases (0.4% of the land surface area)."*

• First Paragraph of the Conclusion :

"Although wetlands are clearly classified based on their hydrological sources, tested methods do not particularly differentiate between freshwater vegetated wetlands (like peatlands and swamps), small streams and saline estuaries. We have avoided double counting of the hydrological roles of wetlands by excluding lakes (Poulter et al., 2017). However, CWs include coastal wetlands, large river corridors and many smaller rivers and river valleys."

**Comment 3 (Validation):**

"Thus the author's wetland area estimate is far higher than most of the existing literature except one recent paper (Hu 2017) which claims 'wetland potential' is~29.8 Mkm2. It is difficult to be convincing that there are > 10 Mkm2 land areas has not been detected and captured by previous studies from satellite observations and inventories unless the authors provide regional validations, especially for the regions that are not covered by previous studies.

A regional validation for Amazonia is needed as this is the largest wetland regions in the world and there are several regional datasets can be used for validation, e.g., work of Melack, Hess and others using PALSAR"

**Reply 3: (Note that this reply is the same for both reviewers' comments regarding validation)**

Our initial manuscript proposed an "evaluation" of regularly flooded wetlands (RFW) and composite wetland (CW) maps at the global scale using three spatial criteria (spatial coincidence, Jaccard Index and spatial Pearson correlation index) compared to three global datasets (out of which two were independent). Both referees have questioned this evaluation and found it insufficient to meet our scientific objectives and not convincing enough to show the added value of the developed wetland maps compared to existing ones. They have both stated that a robust validation is essential.

We completely agree with the necessity of validation and for that, we have revised the structure of the paper by:

(I) Dedicating a whole new section (Sect. 4) to validation and not evaluation.

(II) Enriching the validation section by:

a) Validation over France where the wetlands have been delimited to our knowledge.

b) Validation over the wetland hotspots initially presented in regional analysis (Sect. 5.4 of the initial manuscript now Sect. 4) where we added the Amazon basin and also Southeast Asia.

c) Regions of validation were chosen to display the diversity of wetlands (arid climate: Sudd Swamp; tropical: Amazon basin and Southeast Asia, Boreal: Ob river basin and the Hudson Bay lowlands)

(III) A quantitative analysis (radar charts and Fig. 11, Table 4 and bias) of our developed maps with regard to available wetland datasets, both at global and regional scales: Inventory based (GLWD-3), groundwater modelling- based (Fan et al., 2013; Hu et al., 2017), Satellite imageries (ESA-CCI land cover, GIEMS-D15 and JRC- surface water), France (MPHFM; Berthier et al., 2014) and the Amazon basin (Hess et al., 2015).

(IV) The above validation leads us to select with more precision two of the developed maps as better representatives of the wetland extent at global and regional scale

To do this we organize the section as follows:

- 1) First of all we show that a classical validation over the global scale is not possible since a "suitable reference truth" does not exist among the available wetland datasets.
  - a) Wetland datasets, whether based on inventories (like GLWD-3), satellite imagery (ESA-CCI, GIEMS-D15, JRC-surface water) or based on groundwater modeling (Fan et al. 2013; Hu et al., 2017) are in disagreement at global scale in wet fraction and spatial pattern (Fig.11, Table 4 and regional figures). Therefore we decide to choose the representatives of regional hotspots for which high quality data exists (France and Amazon) to understand where this difference is coming from.
  - b) In the regional scale (Fig. 11) we observe that GLWD-3 systematically underestimates the wetlands: in France (floodplains of the Loire, Saône and Rhône floodplain) and also on the majority of wetland hotspots, like in the Southeast Asia where flooded it represents only a third of flooded wetland. It seems that inventories are exhaustive only over the Hudson Bay lowlands.
  - c) Satellite imagery is not capable of detecting most wetlands (e.g. Hudson Bay lowlands, France) and we can suppose that most of the groundwater-driven wetlands (non-inundated) are not detected by remote sensing as well. These datasets are not seeing areas where no real regional in-situ information for verification exists (e.g. Hess et al., 2015). Identifying wetland from high resolution imagery and photos is not easy and it is not always promising (Friedl et al., 2010; Olofsson et al., 2012).
  - d) Groundwater modelling produces almost always larger wetlands than other datasets that are sometimes in spatial disagreement (Hudson Bay lowland, Ob river basin), less than regularly flooded wetlands (Southeast Asia), or sometimes irrelevant extents and pattern (Sudd swamp in Hu et al. 2017). On the other hand inventories gathered in GLWD-3, overlap of inundation datasets in RFW or a combination of the two (Poulter et al., 2017) are insufficient to represent total wetlands (demonstrated over France by Pison et al., 2018).

- e) As preliminary conclusion, we shows that disagreements at both scales exists among existing datasets, all having their advantages and limitations. The combination of satellite imagery and groundwater modelling can help reduce the disagreements. Although validation is both necessary and challenging at the regional scale, for which France and Amazon are proposed.
- 2) We show that this method works:
  - a) We quantitatively evaluate the CW maps both at global and regional scales (Fig. 4 and Table 4)
  - b) And this lead us to select two CW maps (CW-TCI15%, CW-WTD) with superior performance and reduced disagreement, shown by comparison of wet fraction in Fig. 11 and Table 4 (e.g. CW-TCI15% is pretty close by construction to the wetland extent in Fan et al., 2013 and Hu et al., 2017)
  - c) Regional analysis at the hotspots (including the new regional data over France and Amazon) confirm the proposition with concrete examples.
  - d) We conclude that regularly flooded wetlands and groundwater driven wetlands are necessary to better take into account wetland spatial distribution at all scales. The wet fraction at the global scale will be among high levels.
- 3) We discuss the validation quality
  - a) by explaining the challenges of validation when exhaustive reference datasets do not exist (inconsistencies at different regions of Fig. 11).
  - b) Raising doubts on reported wetland fractions over humid areas like the Amazon mainly driven from overlooking GDWs (large inconsistency among GLWD-3: 8%, Fan et al., 2013: 35%).
  - c) Better performance of CW-WTD over boreal zones since Fan et al. (2013) model explicitly accounts for the permafrost effect on drainage.

**Modifications 3:**

• Two regional validation datasets are added in Sect. 2.5:

**"2.5 Validation datasets**

**2.5.3 Amazon basin wetland map (Hess et al., 2015)**

"Hess et al. (2015) used the L-band SAR data from JRES-1 satellite imagery scenes at a 100m resolution to map wetlands during the period of 1995-1996 for high and low water seasons. The studied domain excludes zones with an altitude higher than 500m and corresponds to a large fraction of the Amazon basin (87%). Wetlands are defined as the sum of lakes, river and other flooded areas plus areas not flooded but adjacent to flooded areas and sharing wetland geomorphology. The flooded fraction of wetlands varies between 38% during the low-water season and 75% during the high-water season with 7% of the wetland area consisting of open water like lakes and rivers (1% of the basin area). The total maximum wetland area which covers 14% of the total basin areas is used for evaluation of CW maps in this study.

**2.5.4 Modelled potentially wet zones of France**

We used a recent national map of potentially wet zones in France (les Milieux Potentiellement Humides de France Modélisée: MPHFM), derived by Berthier et al. (2014) at 50 m resolution based on the topoclimatic soil moisture index (Mérot et al., 2003) and the elevation difference to streams using national high resolution DEMs. Meteorological data for calculation of the topography-climate index (precipitation and potential evaporation rates, see further details in Sect. 3.2.2) are taken from the SAFRAN atmospheric reanalysis (Vidal et al., 2010) at 8 km resolution. The thresholding for wetland delineation is performed independently in 22 hydro-ecoregion units, delimited based on lithology, drainage density, elevation, slope, precipitation rate and temperature. The required percentage of wetlands in each hydroecoregion is taken as the fraction of hydromorphic soils, taken from national soil maps at 1:250,000 (InfoSol, 2013). Additionally, the elevation difference between land pixels and natural streams was used to separate large stream-beds and plain zones which are difficult to model with indices based on topography. Eventually they validated their dataset using available pedological point data (based on profiles or surveys) available over France. These point data are classified into wetlands, non-wetlands and particular cases for the validation procedure. For this procedure they use statistic criteria like gross agreement percentage (number of correctly diagnosed points over total number of points) and Kappa coefficient (modeling error compared to a random classification error)."

• Subsection for France, the Amazon and Southeast Asia validation in new Sect. 4

**"4. Validation**

**France**

Over France (543,000 km2 in metropolitan area excluding Corsica), the wetland fraction is very uncertain based on published global and regional studie , since it ranges between less than 1 to 23% (Fig. 5). The GLWD-3 map shows very few wetlands over France (mostly the Brenne Natural Park in central France, and some coastal wetlands), like the ESA-CCI and JRC datasets. The other global validation datasets (GDW-WTD and Hu et al., 2017) show significantly more wetlands (14 and 18%), scattered over a large fraction of the country apart from mountains, with denser wetlands along large rivers (like the Rhine floodplain at the eastern border) and the Landes (South-Western shore). RFW (with a pattern very similar to GIEMS-D15) has a similar wetland extent (12%) but mostly concentrated as coastal wetlands and in the floodplains of the northern rivers (Loire, Seine, Somme, and Scheldt at the border with Belgium). The MPHFM map, which was tailored for France based on hydromorphic soils by Berthier et al. (2014), shows even larger wetland extents (23% of France), and includes the scattered wetlands of the global validation datasets, most of the RFW, plus dense wetlands over the weakly permeable granites of Brittany (in green on Fig. 5g) and in the floodplains of the Loire (central France), Saône and Rhône floodplains (along 5°E). It can further be noted that almost 42% of the RFWs are reported as wetlands in MPHFM, suggesting the remaining wetlands in MPHFM are mainly groundwater-driven.

These results indicate that the global wetland datasets underestimate wetland extent over France, and very massively for GLWD-3. This conclusion is consistent with the work of Pison et al. (2018), who found that methane emissions over France deduced from the inversion of atmospheric concentrations were much higher (by a third) than estimates based on direct modelling using state-of-the-art global wetland datasets. The two CW maps capture many features of the MPHFM map, including the total wetland extent, although they underestimate wetland density in Brittany, the Landes, and the Saône floodplain. This analysis is confirmed by the much higher similarity criteria when comparing the CW maps to Hu et al. (2017), GDW-WTD or MPHFM than to GLWD-3 (Fig. 4b, Table 4, Table S1 and S2 of the supplementary). Keeping only the independent validation datasets (by excluding GDW-WTD), the CW maps match better the MPHFM dataset, which can be considered as the best validation dataset. It is difficult, however, to identify the best CW map over France based on the similarity criteria against

MPHFM, since CW-WTD, CW-TCI(15%), and CW-TCI(6.6%) display almost the same values (Table S1 of the supplementary)."

**Amazon**

With a basin of 7.5 million  $km^2$  and a mean precipitation rate above 2100 mm/y, the Amazon River has the largest river discharge in the world. We limited our study to the basin used in Hess et al. (2015), which covers 5 million  $km^2$  (Fig. 6). GLWD-3 shows a very similar pattern to RFW suggesting lack of information for noninundated wetlands at regional level in Amazonia rainforests. Similarly, all of the wetland datasets based on remote sensing techniques (i.e. ESA-CCI, JRC surface water, GIEMS-D15 and Hess et al., 2015) show only the main drainage network of the Amazon and parts of the floodplains as wetlands. This contributes to the large difference between the remote sensing based wetlands (maximum: 6% in GIEMS-D15) and those based on GW modelling in Fig. 6 (24 to 35% in Hu et al., 2017 and GDW-WTD). Almost two third of this domain is covered by dense rainforests, which hinder earth surface water observation through both satellite imagery and in-situ measurements. This difference can also be related to the existence of large non-flooded wetlands over Amazons that are not detected through remote sensing. Over the Amazon basin evaluation criteria over Amazon show that RFW is far more similar to Hess et al. (2015) than to other wetland datasets including groundwater-driven wetlands (Fig. 4c). This is clear for example in the Llanos de Moxos wetlands in the lowlands of Bolivia (Southern part of the basin between 12°30' - 17°30' S, 63°-68° W). Assuming the regional dataset of Hess et al. (2015) gives the closest to truth wetland extent, Figure 6 shows that wetlands in the Llanos de Moxos are significantly underestimated in all existing datasets potentially because of their seasonal characteristics or dense herbaceous cover, but this zone is better represented in RFW and CW maps. Large non-inundated wetlands of North-central parts of the basin (delineated by Fan et al., 2013 and Hu et al., 2017) which have not been detected by the remote sensing-based studies are included in CW maps. The added value of CW maps over the Amazon with regard to GW models is its better representation of river channels and surrounding floodplains thanks to RFW component (e.g. downstream Amazon and Tapajós River are not correctly mapped in Hu et al., 2017). Although wetlands of CW maps are more concentrated over the North and North western parts of the basin, they also exist (in lower concentration) over flat South Eastern plains and also in parts of the Andes dry regions (notable in GDW-WTD around 10° S, 75° W).

**South-East Asian deltas**

The window over South and South-East Asia and maritime Southeast Asia chosen in this study for validation stretches over very wet regions where effective annual precipitation at some areas exceeds 3200 mm (e.g. central Bangladesh). The excess water flows downstream and forms wetlands wherever the topography favors water accumulation and wetlands formation. The interest of this regional comparison is its similarity to the Amazon River basin both in the climatic and geographic characters but with severe human interferences and deforestations which has enhanced the satellite remote sensing of the land surface (it is one of the few places on Earth where RFWs are larger than wetlands of validation datasets, Fig. 7d,e,f,g). This region contains several large rivers with vast deltas flowing south towards Bay of Bengal, Andaman Sea, Gulf of Thailand and South China Sea. Among different CW maps, selected CW maps show higher similarities with validation datasets, in particular for SC and SPC criteria (Fig. 4d). The wetland fraction of CW-TCI(15%) and CW-WTD (covering up to 41% of the study window) is almost two times of the maximum wetland extent in evaluation datasets (Fig. 7). Between 58 to 84% of wetlands are correctly detected by the selected CW maps and this criteria is pretty close for RFW map as well assumedly because of the less dense vegetation cover compared to the Amazon (Fig. 4d). Generally

speaking, evaluation criteria are pretty close for RFW and CW maps, meaning that the GDWs are as correctly localized as the RFWs. The spatial correlation between GDW-WTD and RFW over this area is the highest among different regions, suggesting a significant role of groundwater discharge in forming flooded wetlands. Although TCI thresholding in CW-TCI(15%) has the best performance among CW maps based on topographic indices, it should be noted that GDW-TI (Fig. 2b,e) shows a sufficiently good performance as well which suggests a strong role of topography in wetland formation over this region. This is mainly due to very feeble floodplain slope (near zero) over Ganges and Brahmaputra rivers which results in high TI values (e.g. the elevation difference between the main Ganges river corridor and the 200 km wide floodplain surrounding it near Ganges-Brahmaputra merging point is less than 2 meters). The outperforming composite wetland maps frequently extend over larger areas than those in GLWD-3, particularly in major river floodplains and deltas (like Ganges, Brahmaputra, Irawaddy, Mekong, Red river), which were previously delineated with approximate polygons (Fig. 7). Another interesting point is that GDW-WTD and GDW-TCI(15%) show a very similar distribution pattern over Southeast Asian deltas and Indonesia (Fig. 2a,c). Since the TCI does not depend on subsurface properties, an interpretation can be that aroundwater wetland formation is almost completely explained by topography and climate in these areas. Thus, the abundance of precipitation seems to be the primary driver for wetland formation over these deltas, either directly through flood propagation along the main streams or via water transfers by GW.

• Evaluation subsections for Hudson Bay lowlands, Ob river basin and the Sudd swamp are turned into validation subsection.

**Hudson Bay lowlands**

The hydrological system of this region is complex because of the temperature effect on water infiltration and snow melting plus the uncertain interaction between the wetlands and interlaced ponds, lakes, streams and rivers.

There is a systematic contrast between the maps of the inundated zones (maximum wet fraction: 21%) and validation datasets (minimum wet fraction: 49%). The comparison underlines the inability of exclusively using visible range satellite imagery in capturing wetlands (e.g. Landsat images used in JRC surface water, Fig. 8c) in zones with thin vegetation cover but with frequent snow cover. On the other hand, ESA-CCI (Fig. 8a) captures a large parts of the wetlands over the southern shores of the James Bay (probably through the use of the SPOT-VEGETATION datasets to compensate the data gaps of MERIS). CW maps perform better than others, particularly CW-WTD which dominantly wins highest evaluation criteria owing to the extension of dense wetlands toward south of 50°N (Fig. 4e) in particular since the GW model by Fan et al (2013) includes an explicit parametrization of the sea-level boundary condition and the permafrost (adjusted to reproduce the "observed wetland areas" in Northern America, Fan and Miguez-Macho, 2011). Wetlands are 57% larger in CW-WTD than in CW-TCI(15%) with a wide dense strip of wetlands extended from the western sides in Manitoba to eastern shores of James Bay. The RFWs extend over 28% of the area making it one of the highest inundated wetland concentrations globally. Yet, GDWs significantly contribute to both total and inundated wetland extents, as shown by the large overlap between the RFW and GDW-WTD for example. Since CW-WTD agrees well with GLWD-3 (derived from observation inventories), we conclude that over the Hudson Bay lowlands window, GLWD-3 includes a larger range of wetland types, either inundated or not, in contrast to other regions of the world.

**Ob River basin**

The Ob River is the third largest Russian river in western Siberia with a basin extending over  $\sim 3 \times 10^6$  km2. Despite a very large annual variability of inundated area (e.g. Mialon et al., 2005), it is certainly known as one of the largest wetland complex in the world. The Ob River basin is also located in the pan-arctic zones where wetland formation is significantly influenced by the thermic properties. Yet, in parallel to permafrost-related wetlands, wetlands are also effected by fluctuations of the Ob River surface flow. Altogether, comparison of the Ob River basin and Hudson Bay lowlands facilitates evaluating the validity of CW maps in boreal zones and in regions with complex hydrological components. The large wetland fractions (Fig. 9) and the gradual snow melting in the Ob River basin, explains the smooth temporal distribution of the discharge during the flooding period (Grippa et al., 2005). Although wetlands in GIEMS-D15 globally extend over larger fractions of lands, this region is one of the few places (as in the HBL) where wetlands in ESA-CCI are more extensive with a pattern similar to GLWD-3 (Fig. 8 and9). Both in the Ob River basin and HBL permafrost infiltration impedance is a major factor in wetland formation. However in both regions TCTrI-based CW maps fail to surpass others in the validation process. Although the four datasets recognizing the contribution of GW to wetland formation in Fig. 9 (GDW-WTD; Hu et al., 2017 and the two selected CW maps) indicate consistently higher wet fractions than others, the uncertainty in the GDW share in total wetland extent is considerable (the wet fraction solely attributed as GDW ranges from 13% to 29% of the Ob River basin surface area). This uncertainty is both over wetlands densities and spatial pattern (e.g. differences between CW-WTD and CW-TCI(15%) is mainly the southward extension of wetlands in CW-WTD linked to the temperature-based adjustment factor in Fan et al., 2013).

**Sudd swamp**

This large wetland is located in eastern South Sudan, at around 300 m above mean sea level. Recognized as a Ramsar site since 2006, it is the largest freshwater wetland (floodplain and swamp) in the Nile basin (Sutcliffe et al., 2016). Seasonal variations and non-saturated wetlands increase the uncertainty of Sudd swamp extent estimations ranging from 7.2 - 48 103 km2 (Mohamed et al., 2004 and references therein). Since the Sudd is located in a warm semi-arid region, it is of interest for the validation process. Wetlands are often expected to form in regions with wet climate and rather low evaporation rates. Hence, acceptable performance of the CW maps over warm climates assures the global quality of the selected wetland dataset.

Figure 10 shows the wetland distribution and extent in several datasets, which ranges from 1 to 27% of the rectangular window. As a result of this vast difference, evaluation criteria are very small whenever CW maps are compared to an independent validation dataset. Wetlands extent in GLWD-3 and Hu et al. (2017) is almost identical as that of the RFW but with very different spatial patterns. Additionally, wetlands in Hu et al. (2017) are very patchy and show sharp changes of wetland density with what seems as periods of 0.5° which resulted in very poor evaluation criteria between Hu et al. (2017) and CW maps. Owing to this technical barrier, the spatial pattern of wetland in Hu et al. (2017) might not be a good source of validation over the Sudd swamp. In spite of low criteria values for CW maps over the Sudd, selected CW maps are in better accordance with validation datasets (Fig. 4g).. The total wetland fraction is almost equal in CW-TCI(15%) and CW-WTD (between 25 and 27%), but with slightly different spatial patterns which suggests that transmissivity is of little role in wetland formation in this region (similar to Southeast Asia). The CW datasets show high wetland densities in the central floodplain, in

rather good agreement with GLWD-3 and regional estimates of saturated soil (compared to visuals in Mohamed et al., 2004; Mohamed and Savenije, 2014), but the surrounding groundwater wetlands are more scattered in CW-TCI(15%) over local flat valley bottoms. The RFW map (with the most contribution from ESA-CCI) has the most similarities with the GLWD-3 with almost one third of RFW pixels overlapping with wetlands in GLWD-3. RFW map additionally contains the seasonally flooded plains west of the White Nile and South of Bahr-el-Ghazal.

**Comment 4:** Minor The Introduction should not cite Prigent in the discussion of L-band applications of radar – their products are C band or short wavelengths.

**Reply 4:** Thanks for noticing. It will be corrected in the modified manuscript.

**Modification 4:**

• Second paragraph of Introduction :

"Longer wavelengths in the microwave band (e.g. L and C band) penetrate better through the cloud and vegetation layer and provide dynamic observations of inundated zones, usually with a trade-off between high resolution with low revisit or domain extent (Li and Chen, 2005; Hess et al., 2015), and coarse resolution with high revisit up to global coverage (Prigent et al., 2007; Papa et al., 2010; Schroeder et al., 2015; Parrens et al., 2017)."

**Comment 5:** The Hu (2017) map in Figure 10 seems has some artificial barriers. Why is that? Have you considered it in your comparison or validation?

**Reply 5:** We are aware of these sharp changes of wetlands in Hu et al. (2017). We could not find an explanation for these spatial contrast in their manuscript. However we guess that this is caused by using input datasets of coarse resolution in their dataset. The spatial pattern of wetlands in Hu et al. (2017) might not be good for comparison at the regional scale, yet the wet fractions over these regions are comparable for relative bias evaluations.

**Modification 5:**

• Second paragraph of the "Sudd swamp" subsection (Page17, L14-17):

"Additionally, wetlands in Hu et al. (2017) are very patchy and show sharp changes of wetland density with what seems as periods of 0.5° which resulted in very poor evaluation criteria between Hu et al. (2017) and CW maps. Owing to this technical barrier, the spatial pattern of wetland in Hu et al. (2017) might not be a good source of validation over the Sudd swamp."

**Cited references (excluding those already cited in the first manuscript)**

Collins, W. J., Bellouin, N., Doutriaux-Boucher, M., Gedney, N., Halloran, P., Hinton, T., Hughes, J., Jones, C. D., Joshi, M., Liddicoat, S., Martin, G., O'Connor, F., Rae, J., Senior, C., Sitch, S., Totterdell, I., Wiltshire, A. and Woodward, S.: Development and evaluation of an Earth-System model – HadGEM2, Geosci. Model Dev., 4(4), 1051–1075, doi:10.5194/gmd-4-1051-2011, 2011.

Friedl, M.A., Sulla-Menashe, D., Tan, B., Schneider, A., Ramankutty, N., Sibley, A. and Huang, X., 2010. MODIS Collection 5 global land cover: Algorithm refinements and characterization of new datasets. *Remote sensing of Environment*, *114*(1), pp.168-182.

Olofsson, P., Stehman, S.V., Woodcock, C.E., Sulla-Menashe, D., Sibley, A.M., Newell, J.D., Friedl, M.A. and Herold, M., 2012. A global land-cover validation data set, part I: Fundamental design principles. *International Journal of Remote Sensing*, *33*(18), pp.5768-5788.

Pison, I., Berchet, A., Saunois, M., Bousquet, P., Broquet, G., Conil, S., Delmotte, M., Ganesan, A., Laurent, O., Martin, D., O'Doherty, S., Ramonet, M., Spain, T. G., Vermeulen, A., and Yver Kwok, C.: How a European network may help with estimating methane emissions on the French national scale, Atmos. Chem. Phys., 18, 3779-3798, https://doi.org/10.5194/acp-18-3779-2018, 2018.

**Supplementary to the replies to reviewers' comments**

Hydrol. Earth Syst. Sci. Discuss., https://doi.org/10.5194/hess-2018-48

Manuscript under review for journal Hydrological and Earth System Sciences

Authors: Tootchi, A., Jost, A., Ducharne, A.

| Number    | Page |
|-----------|------|
| Table 1   | 2    |
| Table 2   | 3    |
| Table 3   | 3    |
| Table 4   | 4    |
| Figure 1  | 5    |
| Figure 2  | 6    |
| Figure 3  | 7    |
| Figure 4  | 8    |
| Figure 5  | 9    |
| Figure 6  | 10   |
| Figure 7  | 11   |
| Figure 8  | 12   |
| Figure 9  | 13   |
| Figure 10 | 14   |
| Figure 11 | 15   |
| Figure 12 | 16   |
| Figure 13 | 17   |
| Figure 14 | 18   |

**Table 1: Summary of water body, wetland and related proxy maps and datasets from the literature. The wet fractions indicated in % in the last column are those indicated in the reference paper or data description for each study.**

|                                            |                                                       |                                                                                             | Wetland extent             |                |
|--------------------------------------------|-------------------------------------------------------|---------------------------------------------------------------------------------------------|----------------------------|----------------|
| Name and reference                         | Resolution                                            | Type of acquisition                                                                         | (million km 2 ) | % of the land* |
| Maltby and Turner (1983)                   | -                                                     | Based on Russian geographical studies                                                       | 8.6                        | 6.6%           |
| Matthews and Fung (1987)                   | 1 degree                                              | Development from soil, vegetation and inundation maps                                       | 5.3 †           | 4.0%           |
| Mitsch and Gosselink (2000)                | Polygons                                              | Gross estimates, Combination of estimates and maps                                          | ~20†                       | ~15.3%         |
| GLWD-3
Lehner and Döll (2004)           | 30 arcsec ~1km                                        | Compilation of national/international maps                                                  | 8.3 - 10.2 ‡    | 6.2 - 7.6%     |
| GLC2000
Bartholomé and Belward (2005)   | 1 km at Equator                                       | SPOT vegetation mission satellite observations                                              | 4.9                        | 3.4%           |
| GIEMS
Prigent et al. (2007)             | 0.25° ~25km                                           | Multi sensor: AVHRR, SSM/I,
Scatterometer ERS                                            | 2.1 - 5.9                  | 1.4 – 4%       |
| Fan et al. (2013)                          | 30 arcsec ~1km                                        | Groundwater modelling                                                                       | ~19.3†                     | ~17%           |
| GLOWABO
Verpoorter et al. (2014)        | Shapefiles of lakes larger than 0.002 km 2 | Satellite imagery: Landsat and SRTM topography                                              | 5                          | 3.7%           |
| SWAMPS
Schroeder et al. (2015)          | 25 km                                                 | Modeling using multi sensor info:
SSM/I, SSM/S, QuikSCAT, ASCAT                          | 7.7 – 12.5 §    | 5.2 - 8.5%     |
| ESA-CCI land cover
Herold et al. (2015) | 10 arcsec ~300m                                       | Multi sensor: SPOT vegetation, MERIS products                                               | 6.1                        | 4.7%           |
| GIEMS-D15
Fluet-Chouinard et al. (2015) | 15 arcsec ~460m                                       | Multi-sensor: SSM/I, ERS-1, AVHRR,
Downscaled from a 0.25° wetland map                   | 6.5 - 17.3                 | 5.0 - 13.2%    |
| G3WBM
Yamazaki et al. (2015)            | 3 arcsec ~90m                                         | Satellite imagery: Landsat                                                                  | 3.2                        | 2.5%           |
| JRC Surface water
Pekel et al. (2016)   | 1 arcsec ~30m                                         | Satellite imagery: Landsat, including
maximum water extent and interannual
occurrence | 2.8 – 4.4                  | 2.1 - 3.4%     |
| HydroLAKES
Messager et al. (2016)       | Shapefiles of lakes larger than 0.1 km 2   | Multiple inventory compilation
including Canadian hydrographic
dataset and SWBD       | 2.7                        | 1.8%           |
| Hu et al. (2017)                           | 1 km                                                  | Development based on topographic wetness index and land-cover                               | 29.8¶                      | 22.5%          |
| Poulter et al. (2017)                      | 0.5° ~50km                                            | Merging SWAMPS and GLWD-3                                                                   | 10.5                       | 7.1%           |

\* Percentages are those from the corresponding journal article or book. If no mention of percentage coverage exists, the value is calculated by dividing the wetland area by the land surface area excluding Antarctica, the glaciated Greenland and lakes.

 $\ensuremath{^\dagger}$  Excluding Caspian sea and large lakes

‡ Excluding Antarctica, glaciated Greenland, including lakes and Caspian sea. Additionally the range in GLWD is different based on interpretation of fractional wetlands.

§ Excluding large water bodies

¶ Including the Caspian sea

| Table 2: Layers of wetlands constructed in the paper, their definition and the subsection where they are explained. Total |
|---------------------------------------------------------------------------------------------------------------------------|
| land area for wetland percentages excludes lakes, Antarctica and the Greenland ice sheet.                                 |

| Layer                                            |       |        | Definition                                                                   | Wetland
percentage | Explained
in |
|--------------------------------------------------|-------|--------|------------------------------------------------------------------------------|-----------------------|-----------------|
| RFW
(Regularly Flooded Wetlands)       |       | lands) | Union of three inundation datasets (ESA-CCI, GIEMS-D15, JRC surface water)   | 9.7%                  | Sect. 3.1       |
| WTD                                              |       | TD     | Pixels with water table depth less than 20 cm (Fan et al. 2013)              | 15%                   | Sect. 3.2.1     |
| GDW
(Groundwater
Driven
Wetland) | TI    | (6%)   | Pixels with highest Tis, covering 15% of total land when combined with RFW   | 6%                    |                 |
|                                                  |       | (15%)  | Pixels with highest TIs values covering 15% of land                          | 15%                   |                 |
|                                                  | тсі   | (6.6%) | Pixels with highest TCIs, covering 15% of total land when combined with RFW  | 6.6%                  |                 |
|                                                  |       | (15%)  | Pixels with highest TCI values covering 15% of land                          | 15%                   | Sect. 5.2.2     |
|                                                  | TCTrI | (6%)   | Pixels with highest TCTrI, covering 15% of total land when combined with RFW | 6%                    |                 |
|                                                  |       | (15%)  | Pixels with highest TCTrI values covering 15% of land                        | 15%                   |                 |
| CW
(Composite
Wetland)                     | WTD   |        | Union of RFW and GDW-WTD                                                     | 21.1%                 |                 |
|                                                  | ті    | (6%)   | Union of RFW and GDW-TI(6%)                                                  | 15%                   |                 |
|                                                  |       | (15%)  | Union of RFW and GDW-TI(15%)                                                 | 22.2%                 |                 |
|                                                  | тсі   | (6.6%) | Union of RFW and GDW-TCI(6.6%)                                               | 15%                   | Sect. 3.3       |
|                                                  |       | (15%)  | Union of RFW and GDW-TCI(15%)                                                | 21.6%                 |                 |
|                                                  | TCTrI | (6%)   | Union of RFW and GDW-TCTrI(6%)                                               | 15%                   |                 |
|                                                  |       | (15%)  | Union of RFW and GDW-TCTrI(15%)                                              | 22.3%                 |                 |

Table 3: Percent of overlap between GDW and RFW (percent of total pixels).

| Groundwater-driven wetland layer | Intersecting with RFW | Non-intersecting with RFW |
|----------------------------------|-----------------------|---------------------------|
| GDW-TI(6%)                       | 0.7%                  | 5.3%                      |
| GDW-TCI(6.6%)                    | 1.3%                  | 5.3%                      |
| GDW-TCTrl(6%)                    | 0.7%                  | 5.3%                      |
| GDW-TI(15%)                      | 2.5%                  | 12.5%                     |
| GDW-TCI(15%)                     | 3.6%                  | 11.4%                     |
| GDW-TCTrl(15%)                   | 2.4%                  | 12.6%                     |
| GDW-WTD(15%)                     | 3.8%                  | 11.2%                     |

| Dataset name      | ESA-CCI | GIEMS-D15 | JRC surface
water | RFW  | GLWD-3 | GDW-WTD | Hu et al. (2017) |
|-------------------|---------|-----------|----------------------|------|--------|---------|------------------|
| GDW-TI(15%)       | -0.07   | 0.11      | 0.03                 | 0.04 | 0.23   | 0.18    | 0.31             |
| GDW-TCTrl(15%)    | -0.04   | -0.01     | -0.10                | 0.01 | 0.17   | 0.26    | 0.26             |
| GDW-TCI(15%)      | 0.12    | 0.24      | 0.03                 | 0.23 | 0.23   | 0.53    | 0.33             |
| GDW-WTD           | 0.27    | 0.29      | 0.07                 | 0.30 | 0.36   | 1.00    | 0.45             |
| CW-TI(6%)         | 0.56    | 0.59      | 0.44                 | 0.91 | 0.21   | 0.34    | 0.33             |
| CW-TCTrI(6%)      | 0.49    | 0.59      | 0.43                 | 0.78 | 0.24   | 0.43    | 0.40             |
| CW-TCI(6.6%)      | 0.58    | 0.64      | 0.40                 | 0.80 | 0.26   | 0.52    | 0.31             |
| CW-TI(15%)        | 0.63    | 0.60      | 0.28                 | 0.57 | 0.31   | 0.40    | 0.32             |
| CW-TCTrl(15%)     | 0.55    | 0.45      | 0.36                 | 0.51 | 0.32   | 0.38    | 0.28             |
| CW-TCI(15%)       | 0.70    | 0.71      | 0.47                 | 0.69 | 0.28   | 0.58    | 0.35             |
| CW-WTD            | 0.63    | 0.69      | 0.37                 | 0.65 | 0.34   | 0.65    | 0.43             |
| ESA-CCI           | 1.00    | 0.33      | 0.66                 | 0.53 | 0.28   | 0.27    | 0.27             |
| GIEMS-D15         | 0.33    | 1.00      | 0.36                 | 0.67 | 0.26   | 0.29    | 0.20             |
| JRC surface water | 0.66    | 0.36      | 1.00                 | 0.40 | 0.07   | 0.07    | 0.07             |
| RFW               | 0.53    | 0.67      | 0.40                 | 1.00 | 0.38   | 0.30    | 0.22             |
| GLWD-3            | 0.28    | 0.26      | 0.07                 | 0.26 | 1.00   | 0.36    | 0.33             |
| Hu et al. (2017)  | 0.27    | 0.20      | 0.07                 | 0.22 | 0.33   | 0.45    | 1.00             |

 Table 4: Correlation between the developed and reference datasets (wetland fractions in 3 arc-min grid-cells). The highest three values in each column are shown in bold format, and grey cells give the values used in Fig. 4.

---

## Author Comment (AC2) · 17 May 2018

We thank anonymous referee #2 for his/her comments. They will, for sure, help enhance the quality of the work and clarify ambiguities. We will address each block of comments in the following with the comments in black, replies in blue and modifications of the manuscript in green. Since the lack of validation was mentioned in different parts of the referee's report, we gathered all of them together to have a single line of response.

As a preamble, we would like to stress that, to develop the validation of our wetland maps recommended by both reviewers, we profoundly changed the structure of the manuscript. A summary of the major modifications to the manuscript with regard to the referee's comments are listed below:

- **Introduction** (stated and clarified the scientific objectives)
- New section: **Sect. 2.1 Wetland definition**
- **Sect. 2.2 Lakes** (revised to explain the lakes mask)
- **Section 2.5** (added regional validation datasets)
- New section: **Section 3** (containing Sect. 3,4 and 5.1 of the previous manuscript)
- New section: **Section 4** (quantitative global and regional validation, regional scale analysis, including Sect. 5.2 and 5.4 of the previous manuscript)
- New section: **Section 5** (Discussion on the characteristics of composite wetland maps and the role of groundwater-driven wetlands)

All the figures and tables will be provided as a supplementary to this document.

**Comment 1:** "I commend the authors on a great deal of GIS data processing, but I struggled to identify a hypothesis or main insight from this study. This study appears not so much intended as a scientific study but rather a mapping effort. "

**Reply 1:** We apologize for not making our scientific questions clear enough in the submitted manuscript. As it is discussed in the manuscript's introduction, our main scientific hypothesis is that flooded wetland and groundwater driven wetland are complementary to each other and both need to be accounted for to correctly capture the diversity of wetlands. In this regard, wetland mapping is a real scientific objective. It is also a challenging task since it is not a subcategory of remote sensing, but more related to hydrological sciences. It not only involves accurately detecting well-known wetlands, but also "guessing" wetlands presence (absence or fraction) in areas where direct wetland observation is not available. In fact, there are very few studies which consider a comprehensive definition for wetlands of various hydrological types. The effort explained in the manuscript was to develop wetland maps with clear distinction between the frequently inundated ones and those groundwater-driven wetlands which do not necessarily get inundated.

**Modifications:** We propose to explicitly state our scientific question in the introduction and also a whole new subsection (2.1: Wetland definition and general mapping strategy) discussing the purpose of the study and definitions.

- Fifth paragraph of Introduction:
  *"The scientific objective of the present work is to contribute to the long lasting effort for wetland delineation at the global scale since Matthews and Fung (1987). Based on the above analysis, our rationale is that inundated and groundwater-driven wetlands are both important contributions to total wetlands, and both need to be accounted for to realistically capture the wetland patterns and extents. This leads to define wetlands as areas which are persistently saturated or near-saturated, because they are regularly subject to inundation or shallow water tables. Although they share many*

*similar environmental properties because of their wet nature, these two types of wetlands cannot be detected using a single method, since shallow water tables are usually not remotely detectable, while inundated areas have other drivers than groundwater. Thus, we propose to rely on data fusion methods, which have proven advantageous to develop high quality products by merging properties from various datasets, in particular for land cover classification (Fritz and See, 2005; Jung et al., 2006; Schepaschenko et al., 2011; Pérez-Hoyos et al., 2012; Tuanmu and Jetz, 2014) including wetland mapping (Ozesmi & Bauer, 2002; Friedl et al., 2010; Poulter et al., 2017). In this framework, we tested several composite wetland (CW) maps, all constructed at 15 arc-sec resolution by merging two complementary classes of wetlands: (1) the regularly flooded wetlands (RFWs) where surface water can be detected at least once a year through satellite imagery; and (2) the groundwater-driven wetlands (GDWs) that might never get inundated, based on groundwater modelling."*

- ●     2.1. Wetland definition and general mapping strategy

  *"The wetland definition behind the proposed composite maps is focused on hydrological functioning, and is not restricted to vegetated wetlands. We aim at including both seasonal and permanent wetlands, as well as the shallow surface water bodies (including rivers, both permanent and intermittent), since these two types of objects are often hydrologically connected. As a result, the transition between them is not sharp and varies seasonally, so they share many environmental properties, and they are difficult to separate based on observations (either in situ or remote). By including the shallow surface water bodies (in the RFW map), we follow the Ramsar classification, but we depart from it regarding large permanent lakes, which are excluded from the composite wetland maps (section 2.2), since they are very specific water bodies, with distinct hydrology and ecology compared to wetlands and even rivers. The groundwater-driven wetlands, in contrast, can be wet without being ever inundated owing to the presence of a shallow water table. As further discussed in section 3.2, they are defined in this study as areas where the mean annual WTD is less than 20 cm, following similar assumptions in the literature (U.S. Army Corps of Engineers, 1987; Constance et al., 2007; Fan and Miguez-Macho, 2011).*

  *Another feature of the proposed wetland maps is that they are static. As stated in Prigent et al. (2007), they represent therefore the "climatological maximum extent of active wetlands and inundation" (for CW and RFW respectively), i.e. the areas that happen to be saturated or near saturated frequently enough to develop the specific features of wetlands (high soil moisture over a significant part of the year leading to reducing conditions in some horizons, and specific flora and fauna). Potential applications of these static maps are to assign specific hydrological properties or processes to the places identified as wetlands or floodplains. As such, the CW can be seen as the spatial support of a particular "hydrotope" (Gurtz et al., 1999; Hattermann et al., 2004), i.e. the hydrological analog of plant functional types (PFTs) for vegetation properties and processes. Other applications regard the estimation of methane production or denitrification by wetlands, especially if combined to a dynamic modelling of the saturation degree within the wetland fractions. In this particular framework, it must be underlined that the CW maps do not completely solve the so-called "double counting" issue for methane emissions (Poulter et al., 2017), since they include shallow permanent flooded areas and rivers (including the small ones, frequently intermittent), for which no dedicated exhaustive dataset is currently available (Raymond et al 2013; Schneider et al., 2017)."*

**Comment 2:** "Evaluation of the mapped wetlands is done using the maps produced by Lehner & Döll (2004) and Hu et al. (2017). It is not explained why it would be reasonable to put more faith into those mapping efforts than in any of the other, in other words, why they would be a suitable reference 'truth'. If all their data is of better quality, then why not just use it instead?

One possibility is that some of the data used in the Lehner & Döll (2004) mapping are of much higher quality than the candidate data sets, but this is not discussed. If so, then validation and accuracy assessment may be possible for those selected regions where such more accurate mapping is available.

Another possibility would be to create generate a stratified randomised sample of locations in different probability classes and use very high-resolution imagery (e.g. Google Earth) to visually develop a validation data set. This type of validation effort is fairly standard for mapping studies, but it may not always be easy to identify wetlands from high-resolution imagery or even photos.

Using the results from such a validation, you might be able to assign a qualitative weighting to the candidate maps and merge them into a single global map of wetland probability."

**Reply 2:**

The validation procedure does not consider the reference datasets as the true wetland pattern on land (please look at the organization of the validation setion below). We consider each of these validation datasets to carry a part of the truth and for this reason we don not pursue a classical validation.

We do not have access to the regional data gathered in the Lehner & Döll (2004) inventory, though they are not exhaustive globally. Also, as mentioned by the referee, identifying wetland from high resolution imagery and photos is not easy and it is not always promising. It has been demonstrated in previous studies that remote sensing methods (for many technical reasons) and also in-situ observations (by lack of dense enough surveys) tend to underestimate wetland extents in many conditions, like the Amazon and the African equatorial belt datasets (Collins et al., 2011; Gumbricht et al., 2017; Melton et al., 2013), but also in less humid climates because satellite imagery largely overlooks the non-inundated wetlands, as recently reported for France by Pison et al. (2018).

**Comment 3 (Validation):**

"In that case, there should be an independent validation effort to determine the accuracy of the derived product."

"This study does need more robust validation using higher quality wetland mapping.

"In summary, as a data production effort, this manuscript does not help to reduce inconsistencies between existing mapping efforts. Without a thorough validation and accuracy assessment, it does not provide a demonstrable advance."

**Reply 3: (Note that this reply is the same for both reviewers' comments regarding validation)**

Our initial manuscript proposed an "evaluation" of regularly flooded wetlands (RFW) and composite wetland (CW) maps at the global scale using three spatial criteria (spatial coincidence, Jaccard Index and spatial Pearson correlation index) compared to three global datasets (out of which two were

independent). Both referees have questioned this evaluation and found it insufficient to meet our scientific objectives and not convincing enough to show the added value of the developed wetland maps compared to existing ones. They have both stated that a robust validation is essential.

We completely agree with the necessity of validation and for that, we have revised the structure of the paper by:

(I) Dedicating a whole new section (Sect. 4) to validation and not evaluation.

(II) Enriching the validation section by:

> a) Validation over France where the wetlands have been delimited to our knowledge.
> b) Validation over the wetland hotspots initially presented in regional analysis (Sect. 5.4 of the initial manuscript now Sect. 4) where we added the Amazon basin and also Southeast Asia.
> c) These regions of validation were chosen to display the diversity of wetlands (arid climate: Sudd Swamp; tropical: Amazon basin and Southeast Asia, Boreal: Ob river basin and the Hudson Bay lowlands)

(III) A quantitative analysis (radar charts and Fig. 11, Table 4 and bias) of our developed maps with regard to available wetland datasets, both at global and regional scales: Inventory based (GLWD-3), groundwater modelling- based (Fan et al., 2013; Hu et al., 2017), Satellite imageries (ESA-CCI land cover, GIEMS-D15 and JRC- surface water), France (MPHFM; Berthier et al., 2014) and the Amazon basin (Hess et al., 2015).

(IV) The above validation leads us to select with more precision two of the developed maps as better representatives of the wetland extent at global and regional scale

To do this we organize the section as follows:

1) First of all we show that a classical validation over the global scale is not possible since a "suitable reference truth" does not exist among the available wetland datasets.
   a) Wetland datasets, whether based on inventories (like GLWD-3), satellite imagery (ESA-CCI, GIEMS-D15, JRC-surface water) or based on groundwater modeling (Fan et al. 2013; Hu et al., 2017) are in disagreement at global scale in wet fraction and spatial pattern (Fig.11, Table 4 and regional figures). Therefore we decide to choose the representatives of regional hotspots for which high quality data exists (France and Amazon) to understand where this difference is coming from.
   b) In the regional scale (Fig. 11) we observe that GLWD-3 systematically underestimates the wetlands: in France (floodplains of the Loire, Saône and Rhône floodplain) and also on the majority of wetland hotspots, like in the Southeast Asia where flooded it represents only a third of flooded wetland. It seems that inventories are exhaustive only over the Hudson Bay lowlands.
   c) Satellite imagery is not capable of detecting most wetlands (e.g. Hudson Bay lowlands, France) and we can suppose that most of the groundwater-driven wetlands (non-inundated) are not detected by remote sensing as well. These datasets are not seeing areas where no real regional in-situ information for verification exists (e.g. Hess et al., 2015). As mentioned by the referee, identifying wetland from high resolution imagery and photos is not easy and it is not always promising (Friedl et al., 2010; Olofsson et al., 2012). On the other hand, we do not have access to the regional data gathered in the Lehner & Döll (2004) inventory.

d)  Groundwater modelling produces almost always larger wetlands than other datasets that are sometimes in spatial disagreement (Hudson Bay lowland, Ob river basin), less than regularly flooded wetlands (Southeast Asia), or sometimes irrelevant extents and pattern (Sudd swamp in Hu et al. 2017). On the other hand inventories gathered in GLWD-3, overlap of inundation datasets in RFW or a combination of the two (Poulter et al., 2017) are insufficient to represent total wetlands (demonstrated over France by Pison et al., 2018).

e)  As preliminary conclusion, we shows that disagreements at both scales exists among existing datasets, all having their advantages and limitations. The combination of satellite imagery and groundwater modelling can help reduce the disagreements. Although validation is both necessary and challenging a the regional scale, for which France and Amazon are proposed.

2)  We show that this method works:

a)  We quantitatively evaluate the CW maps both at global and regional scales (Fig. 4 and Table 4)

b)  And this lead us to select two CW maps (CW-TCI15%, CW-WTD) with superior performance and reduced disagreement, shown by comparison of wet fraction in Fig. 11 and Table 4 (e.g. CW-TCI15% is similar by construction to the wetland extent in Fan et al., 2013 and Hu et al., 2017)

c)  Regional analysis at the hotspots (including the new regional data over France and Amazon) confirm the proposition with concrete examples.

d)  We conclude that regularly flooded wetlands and groundwater driven wetlands are necessary to better take into account wetland spatial distribution at all scales. The wet fraction at the global scale will be among high levels.

3)  We discuss the validation quality

a)  By explaining the challenges of validation when exhaustive reference datasets do not exist (inconsistencies at different regions of Fig. 11).

b)  Raising doubts on reported wetland fractions over humid areas like the Amazon mainly driven from overlooking GDWs (large inconsistency among GLWD-3: 8%, Fan et al., 2013: 35%).

c)  Better performance of CW-WTD over boreal zones since Fan et al. (2013) model explicitly accounts for the permafrost effect on drainage.

**Modifications 3:**

● Two regional validation datasets are added in Sect. 2.5:

*"2.5 Validation datasets*

*2.5.3 Amazon basin wetland map (Hess et al., 2015)*

*"Hess et al. (2015) used the L-band SAR data from JRES-1 satellite imagery scenes at a 100m resolution to map wetlands during the period of 1995-1996 for high and low water seasons. The studied domain excludes zones with an altitude higher than 500m and corresponds to a large fraction of the Amazon basin (87%). Wetlands are defined as the sum of lakes, river and other flooded areas plus areas not flooded but adjacent to flooded areas and sharing wetland geomorphology. The flooded fraction of wetlands varies between 38% during the low-water season and 75% during the high-water season with 7% of the wetland area consisting of open water like lakes and rivers (1% of the basin area). The total maximum wetland area which covers 14% of the total basin areas is used for evaluation of CW maps in this study.*

***2.5.4 Modelled potentially wet zones of France***

*We used a recent national map of potentially wet zones in France (les Milieux Potentiellement Humides de France Modélisée: MPHFM), derived by Berthier et al. (2014) at 50 m resolution based on the topo-climatic soil moisture index (Mérot et al., 2003) and the elevation difference to streams using national high resolution DEMs. Meteorological data for calculation of the topography-climate index (precipitation and potential evaporation rates, see further details in Sect. 3.2.2) are taken from the SAFRAN atmospheric reanalysis (Vidal et al., 2010) at 8 km resolution. The thresholding for wetland delineation is performed independently in 22 hydro-ecoregion units, delimited based on lithology, drainage density, elevation, slope, precipitation rate and temperature. The required percentage of wetlands in each hydro-ecoregion is taken as the fraction of hydromorphic soils, taken from national soil maps at 1:250,000 (InfoSol, 2013). Additionally, the elevation difference between land pixels and natural streams was used to separate large stream-beds and plain zones which are difficult to model with indices based on topography. Eventually they validated their dataset using available pedological point data (based on profiles or surveys) available over France. These point data are classified into wetlands, non-wetlands and particular cases for the validation procedure. For this procedure they use statistic criteria like gross agreement percentage (number of correctly diagnosed points over total number of points) and Kappa coefficient (modeling error compared to a random classification error)."*

- Subsection for France, the Amazon and SouthEast Asia validation in new Sect. 4

*"**4. Validation***

***France***

*Over France (543,000 km$^2$ in metropolitan area excluding Corsica), the wetland fraction is very uncertain based on published global and regional studies, since it ranges between less than 1 to 23% (Fig. 5). The GLWD-3 map shows very few wetlands over France (mostly the Brenne Natural Park in central France, and some coastal wetlands), like the ESA-CCI and JRC datasets. The other global validation datasets (GDW-WTD and Hu et al., 2017) show significantly more wetlands (14 and 18%), scattered over a large fraction of the country apart from mountains, with denser wetlands along large rivers (like the Rhine floodplain at the eastern border) and the Landes (South-Western shore). RFW (with a pattern very similar to GIEMS-D15) has a similar wetland extent (12%) but mostly concentrated as coastal wetlands and in the floodplains of the northern rivers (Loire, Seine, Somme, and Scheldt at the border with Belgium). The MPHFM map, which was tailored for France based on hydromorphic soils by Berthier et al. (2014), shows even larger wetland extents (23% of France), and includes the scattered wetlands of the global validation datasets, most of the RFW, plus dense wetlands over the weakly permeable granites of Brittany (in green on Fig. 5g) and in the floodplains of the Loire (central France), Saône and Rhône floodplains (along 5°E). It can further be noted that almost 42% of the RFWs are reported as wetlands in MPHFM, suggesting the remaining wetlands in MPHFM are mainly groundwater-driven.*

*These results indicate that the global wetland datasets underestimate wetland extent over France, and very massively for GLWD-3. This conclusion is consistent with the work of Pison et al. (2018), who found that methane emissions over France deduced from the inversion of atmospheric concentrations were much higher (by a third) than estimates based on direct modelling using state-of-the-art global wetland datasets. The two CW maps capture many features of the MPHFM map, including the total wetland extent, although they underestimate wetland density in Brittany, the Landes, and the Saône floodplain. This analysis is confirmed by the much higher similarity criteria when comparing the CW maps to Hu et al. (2017), GDW-WTD or MPHFM than to GLWD-3 (Fig. 4b, Table 4, Table S1 and S2 of the supplementary). Keeping only the independent validation datasets (by excluding GDW-WTD), the*

*CW maps match better the MPHFM dataset, which can be considered as the best validation dataset. It is difficult, however, to identify the best CW map over France based on the similarity criteria against MPHFM, since CW-WTD, CW-TCI(15%), and CW-TCI(6.6%) display almost the same values (Table S1 of the supplementary)."*

**Amazon**

*With a basin of 7.5 million $km^2$ and a mean precipitation rate above 2100 mm/y, the Amazon River has the largest river discharge in the world. We limited our study to the basin used in Hess et al. (2015), which covers 5 million $km^2$ (Fig. 6). GLWD-3 shows a very similar pattern to RFW suggesting lack of information for non-inundated wetlands at regional level in Amazonia rainforests. Similarly, all of the wetland datasets based on remote sensing techniques (i.e. ESA-CCI, JRC surface water, GIEMS-D15 and Hess et al., 2015) show only the main drainage network of the Amazon and parts of the floodplains as wetlands. This contributes to the large difference between the remote sensing based wetlands (maximum: 6% in GIEMS-D15) and those based on GW modelling in Fig. 6 (24 to 35% in Hu et al., 2017 and GDW-WTD). Almost two third of this domain is covered by dense rainforests, which hinder earth surface water observation through both satellite imagery and in-situ measurements. This difference can also be related to the existence of large non-flooded wetlands over Amazons that are not detected through remote sensing. Over the Amazon basin evaluation criteria over Amazon show that RFW is far more similar to Hess et al. (2015) than to other wetland datasets including groundwater-driven wetlands (Fig. 4c). This is clear for example in the Llanos de Moxos wetlands in the lowlands of Bolivia (Southern part of the basin between 12°30' - 17°30' S, 63°-68° W). Assuming the regional dataset of Hess et al. (2015) gives the closest to truth wetland extent, Figure 6 shows that wetlands in the Llanos de Moxos are significantly underestimated in all existing datasets potentially because of their seasonal characteristics or dense herbaceous cover, but this zone is better represented in RFW and CW maps. Large non-inundated wetlands of North-central parts of the basin (delineated by Fan et al., 2013 and Hu et al., 2017) which have not been detected by the remote sensing-based studies are included in CW maps. The added value of CW maps over the Amazon with regard to GW models is its better representation of river channels and surrounding floodplains thanks to RFW component (e.g. downstream Amazon and Tapajós River are not correctly mapped in Hu et al., 2017). Although wetlands of CW maps are more concentrated over the North and North western parts of the basin, they also exist (in lower concentration) over flat South Eastern plains and also in parts of the Andes dry regions (notable in GDW-WTD around 10° S, 75° W).*

**South-East Asian deltas**

*The window over South and South-East Asia and maritime Southeast Asia chosen in this study for validation stretches over very wet regions where effective annual precipitation at some areas exceeds 3200 mm (e.g. central Bangladesh). The excess water flows downstream and forms wetlands wherever the topography favors water accumulation and wetlands formation. The interest of this regional comparison is its similarity to the Amazon River basin both in the climatic and geographic characters but with severe human interferences and deforestations which has enhanced the satellite remote sensing of the land surface (it is one of the few places on Earth where RFWs are larger than wetlands of validation datasets, Fig. 7d,e,f,g). This region contains several large rivers with vast deltas flowing south towards Bay of Bengal, Andaman Sea, Gulf of Thailand and South China Sea. Among different CW maps, selected CW maps show higher similarities with validation datasets, in particular for SC and SPC criteria (Fig. 4d). The wetland fraction of CW-TCI(15%) and CW-WTD (covering up to 41% of the study window) is almost two times of the maximum wetland extent in evaluation datasets (Fig. 7). Between 58 to 84% of wetlands are correctly detected by the selected CW maps and this*

*criteria is pretty close for RFW map as well assumedly because of the less dense vegetation cover compared to the Amazon (Fig. 4d). Generally speaking, evaluation criteria are pretty close for RFW and CW maps, meaning that the GDWs are as correctly localized as the RFWs. The spatial correlation between GDW-WTD and RFW over this area is the highest among different regions, suggesting a significant role of groundwater discharge in forming flooded wetlands. Although TCI thresholding in CW-TCI(15%) has the best performance among CW maps based on topographic indices, it should be noted that GDW-TI (Fig. 2b,e) shows a sufficiently good performance as well which suggests a strong role of topography in wetland formation over this region. This is mainly due to very feeble floodplain slope (near zero) over Ganges and Brahmaputra rivers which results in high TI values (e.g. the elevation difference between the main Ganges river corridor and the 200 km wide floodplain surrounding it near Ganges-Brahmaputra merging point is less than 2 meters). The outperforming composite wetland maps frequently extend over larger areas than those in GLWD-3, particularly in major river floodplains and deltas (like Ganges, Brahmaputra, Irawaddy, Mekong, Red river), which were previously delineated with approximate polygons (Fig. 7). Another interesting point is that GDW-WTD and GDW-TCI(15%) show a very similar distribution pattern over Southeast Asian deltas and Indonesia (Fig. 2a,c). Since the TCI does not depend on subsurface properties, an interpretation can be that groundwater wetland formation is almost completely explained by topography and climate in these areas. Thus, the abundancy of precipitation seems to be the primary driver for wetland formation over these deltas, either directly through flood propagation along the main streams or via water transfers by GW.*

- *Evaluation subsections for Hudson Bay lowlands, Ob river basin and the Sudd swamp are turned into validation subsection.*

**Hudson Bay lowlands**

*The hydrological system of this region is complex because of the temperature effect on water infiltration and snow melting plus the uncertain interaction between the wetlands and interlaced ponds, lakes, streams and rivers.*

*There is a systematic contrast between the maps of the inundated zones (maximum wet fraction: 21%) and validation datasets (minimum wet fraction: 49%). The comparison underlines the inability of exclusively using visible range satellite imagery in capturing wetlands (e.g. Landsat images used in JRC surface water, Fig. 8c) in zones with thin vegetation cover but with frequent snow cover. On the other hand, ESA-CCI (Fig. 8a) captures a large parts of the wetlands over the southern shores of the James Bay (probably through the use of the SPOT-VEGETATION datasets to compensate the data gaps of MERIS). CW maps perform better than others, particularly CW-WTD which dominantly wins highest evaluation criteria owing to the extension of dense wetlands toward south of 50°N (Fig. 4e) in particular since the GW model by Fan et al (2013) includes an explicit parametrization of the sea-level boundary condition and the permafrost (adjusted to reproduce the "observed wetland areas" in Northern America, Fan and Miguez-Macho, 2011). Wetlands are 57% larger in CW-WTD than in CW-TCI(15%) with a wide dense strip of wetlands extended from the western sides in Manitoba to eastern shores of James Bay. The RFWs extend over 28% of the area making it one of the highest inundated wetland concentrations globally. Yet, GDWs significantly contribute to both total and inundated wetland extents, as shown by the large overlap between the RFW and GDW-WTD for example. Since CW-WTD agrees well with GLWD-3 (derived from observation inventories), we conclude that over the Hudson Bay lowlands window, GLWD-3 includes a larger range of wetland types, either inundated or not, in contrast to other regions of the world.*

**Ob River basin**

*The Ob River is the third largest Russian river in western Siberia with a basin extending over ~3×10$^6$ km$^2$. Despite a very large annual variability of inundated area (e.g. Mialon et al., 2005), it is certainly known as one of the largest wetland complex in the world. The Ob River basin is also located in the pan-arctic zones where wetland formation is significantly influenced by the thermic properties. Yet, in parallel to permafrost-related wetlands, wetlands are also effected by fluctuations of the Ob River surface flow. Altogether, comparison of the Ob River basin and Hudson Bay lowlands facilitates evaluating the validity of CW maps in boreal zones and in regions with complex hydrological components. The large wetland fractions (Fig. 9) and the gradual snow melting in the Ob River basin, explains the smooth temporal distribution of the discharge during the flooding period (Grippa et al., 2005). Although wetlands in GIEMS-D15 globally extend over larger fractions of lands, this region is one of the few places (as in the HBL) where wetlands in ESA-CCI are more extensive with a pattern similar to GLWD-3 (Fig. 8 and9). Both in the Ob River basin and HBL permafrost infiltration impedance is a major factor in wetland formation. However in both regions TCTrI-based CW maps fail to surpass others in the validation process. Although the four datasets recognizing the contribution of GW to wetland formation in Fig. 9 (GDW-WTD; Hu et al., 2017 and the two selected CW maps) indicate consistently higher wet fractions than others, the uncertainty in the GDW share in total wetland extent is considerable (the wet fraction solely attributed as GDW ranges from 13% to 29% of the Ob River basin surface area). This uncertainty is both over wetlands densities and spatial pattern (e.g. differences between CW-WTD and CW-TCI(15%) is mainly the southward extension of wetlands in CW-WTD linked to the temperature-based adjustment factor in Fan et al., 2013).*

**Sudd swamp**

*This large wetland is located in eastern South Sudan, at around 300 m above mean sea level. Recognized as a Ramsar site since 2006, it is the largest freshwater wetland (floodplain and swamp) in the Nile basin (Sutcliffe et al., 2016). Seasonal variations and non-saturated wetlands increase the uncertainty of Sudd swamp extent estimations ranging from 7.2 - 48 10$^3$ km$^2$ (Mohamed et al., 2004 and references therein). Since the Sudd is located in a warm semi-arid region, it is of interest for the validation process. Wetlands are often expected to form in regions with wet climate and rather low evaporation rates. Hence, acceptable performance of the CW maps over warm climates assures the global quality of the selected wetland dataset.*

*Figure 10 shows the wetland distribution and extent in several datasets, which ranges from 1 to 27% of the rectangular window. As a result of this vast difference, evaluation criteria are very small whenever CW maps are compared to an independent validation dataset. Wetlands extent in GLWD-3 and Hu et al. (2017) is almost identical as that of the RFW but with very different spatial patterns. Additionally, wetlands in Hu et al. (2017) are very patchy and show sharp changes of wetland density with what seems as periods of 0.5° which resulted in very poor evaluation criteria between Hu et al. (2017) and CW maps. Owing to this technical barrier, the spatial pattern of wetland in Hu et al. (2017) might not be a good source of validation over the Sudd swamp. In spite of low criteria values for CW maps over the Sudd, selected CW maps are in better accordance with validation datasets (Fig. 4g).. The total wetland fraction is almost equal in CW-TCI(15%) and CW-WTD (between 25 and 27%), but with slightly different spatial patterns which suggests that transmissivity is of little role in wetland formation in this region (similar to Southeast Asia). The CW datasets show high wetland densities in the central floodplain, in rather good agreement with GLWD-3 and regional estimates of saturated soil (compared to visuals in Mohamed et al., 2004; Mohamed and Savenije, 2014), but the surrounding groundwater wetlands are more scattered in CW-TCI(15%) over local flat valley bottoms. The RFW map (with the most contribution from ESA-CCI) has the most similarities with the GLWD-3*

*with almost one third of RFW pixels overlapping with wetlands in GLWD-3. RFW map additionally contains the seasonally flooded plains west of the White Nile and South of Bahr-el-Ghazal."*

**Comment 4:** I don't understand the meaning of the word 'scattered' groundwater wetlands. P3, l38 suggests the two classes are complementary yet intersect. This cannot both be correct in the formal sense, and indeed they are not complementary. What about irregularly flooded wetlands, flooded groundwater wetlands, contiguous groundwater wetlands, scattered flooded wetlands, etc? In other words, the conceptual classification framework needs more thought. A broader distinction between surface water and groundwater-dominated wetlands might work better, for example.

**Reply 4:** We agree that the use of the word "scattered" is ambiguous in the context of this study since not all groundwater-driven wetlands are scattered. We have revised the naming protocol in the modified version accordingly: what we called Scattered Groundwater Wetlands (SGW) is now noted as Groundwater-Driven Wetlands (GDW).

Our classification objective is to divide wetlands into those often affected by surface water processes and those fed through groundwater convergence. Also these names be related to their determination protocol (satellite/GW modelling) and their properties. This classification is not all-inclusive. Regularly flooded wetlands (RFW) can also have groundwater sources. Irregularly flooded wetlands are not often detected by satellite imageries because of their temporal nature.

**Modifications 4:** The naming is modified many places within the text in the new manuscript. About the intersection please refer to Table 3.

**Comment 5:** The language needs more work. It is sometimes incorrect, sometimes imprecise or ambiguous. Incorrectly used or imprecise words used include: massive, detecting, pretty, players, popular, believed, replica, and patches. The grammar is also lacking in places and needs checking (e.g., "in the high end", "In latter", "Back to").

**Reply 5:** Improper vocabulary as indicated by the reviewer has been detected along the manuscript and modified. Regarding the grammar and improper vocabulary, we will get the manuscript corrected by a professional English editing service.

**Comment 6:** P4,l18 – Please describe what data sets there are and how you know that they are not significantly different.

**Reply 6:** We were not clear with regard to our choice of lakes mask in the first manuscript. We completely modified the lakes sub-section with a more clear description of the lakes mask. The whole paragraph is inserted in the modification part below.

**Modifications 6:**

- 2.2 Lakes:
*"To distinguish large permanent lakes from wetlands, we used the HydroLAKES database (Messager et al., 2016), which was developed by compiling national, regional and global datasets. It consists of more than 1.4 million individual polygons for lakes with a surface area of at least 10 ha, covering 1.8% of the land surface area. This value is smaller than in other recent databases accounting for smaller water bodies: 2.5% in G3WBM (Yamazaki et al., 2015) for water bodies above 0.8 ha; 3.5% in GLOWABO (Verpoorter et al., 2014) for those above 0.2 ha. These two datasets do not differentiate lakes, and using them as a lake mask would lead to exclude areas that can be considered as wetlands according to Sect 2.1, like shallow water bodies in important wetland sites, like in the Ob*

*river basin, Indonesian mangroves, or Ganges floodplains. These areas can be recognized as wetlands by the datasets underlying the RFW map (Sect. 2.3), so we based the lake mask on HydroLAKES to conserve these wetlands. It must also be noted that the small water bodies tend to be overlooked after dominant resampling to the 15 arc-sec (Sect 2.6), unless they are sufficiently numerous in a pixel. This explains why the lake mask shown in Fig. 1a covers only 1.7% of the land area, compared to 1.8% in the original HydroLAKES database. This map also shows that most of the lakes are located in the northern boreal zones (more than 60% of lakes area is north 50°N), in agreement with the other lake databases."*

**Comment 7:-** p4,l30 - Pls describe what method is used to delineate wetlands in the ESA-CCI product.

**Reply 7:** The permanent water bodies are delineated using MERIS reflection data and the regularly flooded regions are delineated by mixed use of vegetation classes and reflection signals. Land cover change detection is performed at the coarser 1 km spatial resolution, based on the AVHRR, SPOT-VGT and PROBA-V mission.

**Modifications 7:** The above mentioned explanation is added to Sect. 2.3.1 Page 5.

*"This dataset succeeds the GlobCover dataset based on the data from MERIS sensor (on board of ENVISAT) at high resolution for surface water detection, along with SPOT-VEGETATION time series (Herold et al., 2015) to help distinguish wetlands from other vegetation covers. Global land cover maps at approximately 300 m (10 arc-sec) resolution deliver data for three 5-year periods (1998-2002, 2003-2007 and 2008-2012). Extent of water bodies have slightly changed between the first 5 year period to the third one (like shrinking of Aral lake by more than 55% of the area), but the extent of wetland classes (permanent wetlands and flooded vegetation classes) did not significantly change (the variation in wetland classes throughout these periods is less than 3% of total wetlands area). We acquired the last epoch data to represent the present state of wetlands (Fig. 1b). In this land cover dataset, legend entries that could be considered as wetlands are mixed classes of flooded areas with tree covers, shrubs or herbaceous covers plus inland water bodies. The flooded classes cover 3% of the Earth land surface."*

**Comment 8:** p4,l35 and further on – journal may not accept a reference in a section header, even less so if you don't repeat it in the main text.

**Reply 8:** There is no explicit suggestion for this limitation in the guidelines for authors. If the journal confirms it, we will change the incriminated headers.

**Comment 9:** p9,l34 – sound circular to me, pls explain if it isn't.

**Reply 9:** GDW identification based on soil moisture indices like TI requires two kinds of information: (1) patterns based on TI maps; (2) a threshold telling which TIs are high enough to actually be good indicators of wetland presence. The same is true for GDW identification based on WTD modelling. As indicated in the fourth paragraph of the Introduction, threshold selection is a major difficulty for GDW delineation, well recognized by hydrologists, and conveying some subjectivity. To explore the uncertainties related to this choice, we relied on a published GDW extent (15% from Fan et al., 2013) and compared to the maps obtained if either the GWD-TI or the CW-TI cover 15% of the land area. In the latter case, the GDW cover less than 15%, between 6 and 6.6% depending on the TI formulation. The conclusions drawn from this analysis are summarized in the 2nd paragraph of the Conclusion.

**Modifications 9:** For clarity, we propose changes in the Introduction

*Introduction*, *Paragraph 4:*

*"A major challenge to identify wetlands, based on either simplified or direct WTD modelling, is to define thresholds, on TI or WTD respectively, so that all pixels having a higher TI or smaller WTD than these thresholds are accounted as wetlands. TI thresholds are often calibrated to reproduce the pattern of documented wetlands in a certain region and then extrapolated for larger domains. This strategy was proven successful at the basin scale (e.g. Curie et al., 2007), but it has been shown ineffective at larger scales, since it is not possible to uniquely link TI values to soil saturation levels across different landforms and climates (Marthews et al., 2015). Consequently, Hu et al. (2017) produced a global wetland dataset by calibrating independent TI thresholds in all the large river basins of the world, as pioneered over France owing to TI threshold calibration in 22 hydro-ecoregions (Berthier et al., 2014). Direct WTD modelling, at continental/global scale, is quite demanding in terms of data and CPU requirements, and has rarely been used for wetland delineation. To our knowledge, the only example is the work of Fan and Miguez-Macho (2011) and Fan et al. (2013), who applied uniform WTD thresholds, first over North America, then globally. They propose WTD thresholds between 0 to 25 cm for wetlands and inundated areas. The resulting wetland patterns are found to be very similar for different thresholds within this range. It must be emphasized that adjusting wetland thresholds, both for directly modelled WTD and TI, always implies subjective choices. "*

- We reshaped the paragraph including the mentioned sentence to underline the uncertainties of TI threshold selection, and the way we tried to explore them within the bounds provided by the global wetland fraction of Fan et al. (2013):

*"Two index thresholds for two global GDW fractions"*

*"Like in many studies (Rodhe and Seibert 1999; Curie et al., 2007; Hu et al., 2017), we define TI-based wetlands as the pixels with TI above a certain threshold, itself defined to match a certain fraction of total land. In doing so, we prescribe the global GDW fraction to a chosen value, and the various TI formulations only change the geographic distribution of the corresponding wetlands. To apprehend the uncertainty related to the choice of the global GDW fraction, we tested two choices within the bounds provided by the global wetland extent of Fan et al (2013), which is by construction the global GDW-WTD extent (Sect. 3.2.1). [DA1] In the first approach, we set the TI threshold so that the wet pixels (with high index values) cover 15% of the land surface area, like the fraction of WTD ≤ 20 cm according to Fan et al. (2013). The corresponding maps are noted as GDW-TI(15%), GDW-TCI(15%) and GDW-TCTrI(15%) in Table 2, and show fairly different patterns (Fig. 2b,c,d). The second approach assumes that the total wetland extent, this time including both GDW and RFW, covers 15%; the TI thresholds are then set so that the union of RFW and GDW-TI (TCI/TCTrI), i.e. the composite wetlands, have the same extent as GDW-WTD."*

- We suggest to add the following paragraph in the conclusion

*Conclusion* (second paragraph)

*"Whether derived from simplified or direct WTD modelling (based on the topographic index, TI, or on the estimates from Fan et al., 2013), a major challenge is to define thresholds on TI or WTD to separate the wet and non-wet pixels. In line with the existing literature, we chose to define wetlands as areas where the mean WTD is less than 20 cm, and this WTD threshold was translated into the TI threshold defining the same global wetland extent (15%). These choices necessarily remain subjective*

*in absence of a consensual global wetland map and definition, and the related uncertainty on wetland extent was shown to amount to a few percents of total land area, based on sensitivity analyses for reasonable values of the different thresholds. We also considered several classical variants of the TI to conclude that the TCI (topography-climate index), also favored by Hu et al. (2017) with a modified formula, was offering the best correspondence with the validation datasets. The original TI did not capture the wetland density contrasts between arid and wet areas, while the inclusion of sub-surface transmissivity in TCTrI induced too sharp density contrasts, not always matching the recognized patterns of large wetlands. This calls for improved global transmissivity datasets, or new methods to provide a more continuous description of transmissivity than what is currently proposed based on discrete classes of lithology (Hartmann and Moosdorf, 2012; Gleeson et al., 2014) or soil texture (Fan et al., 2013)."*

**Cited references (excluding those already cited in the first manuscript)**

Collins, W. J., Bellouin, N., Doutriaux-Boucher, M., Gedney, N., Halloran, P., Hinton, T., Hughes, J., Jones, C. D., Joshi, M., Liddicoat, S., Martin, G., O'Connor, F., Rae, J., Senior, C., Sitch, S., Totterdell, I., Wiltshire, A. and Woodward, S.: Development and evaluation of an Earth-System model – HadGEM2, Geosci. Model Dev., 4(4), 1051–1075, doi:10.5194/gmd-4-1051-2011, 2011.

Friedl, M.A., Sulla-Menashe, D., Tan, B., Schneider, A., Ramankutty, N., Sibley, A. and Huang, X., 2010. MODIS Collection 5 global land cover: Algorithm refinements and characterization of new datasets. *Remote sensing of Environment*, *114*(1), pp.168-182.

Olofsson, P., Stehman, S.V., Woodcock, C.E., Sulla-Menashe, D., Sibley, A.M., Newell, J.D., Friedl, M.A. and Herold, M., 2012. A global land-cover validation data set, part I: Fundamental design principles. *International Journal of Remote Sensing*, *33*(18), pp.5768-5788.

Pison, I., Berchet, A., Saunois, M., Bousquet, P., Broquet, G., Conil, S., Delmotte, M., Ganesan, A., Laurent, O., Martin, D., O'Doherty, S., Ramonet, M., Spain, T. G., Vermeulen, A., and Yver Kwok, C.: How a European network may help with estimating methane emissions on the French national scale, Atmos. Chem. Phys., 18, 3779-3798, https://doi.org/10.5194/acp-18-3779-2018, 2018.

**Supplementary to the replies to reviewers' comments**

Authors: Tootchi, A., Jost, A., Ducharne, A.

**Table 1: Summary of water body, wetland and related proxy maps and datasets from the literature. The wet fractions indicated in % in the last column are those indicated in the reference paper or data description for each study.**

| Name and reference | Resolution | Type of acquisition | Wetland extent (million km²) | % of the land* |
|---|---|---|---|---|
| Maltby and Turner (1983) | - | Based on Russian geographical studies | 8.6 | 6.6% |
| Matthews and Fung (1987) | 1 degree | Development from soil, vegetation and inundation maps | 5.3[†] | 4.0% |
| Mitsch and Gosselink (2000) | Polygons | Gross estimates, Combination of estimates and maps | ~20[†] | ~15.3% |
| GLWD-3 Lehner and Döll (2004) | 30 arcsec ~1km | Compilation of national/international maps | 8.3 - 10.2[‡] | 6.2 - 7.6% |
| GLC2000 Bartholomé and Belward (2005) | 1 km at Equator | SPOT vegetation mission satellite observations | 4.9 | 3.4% |
| GIEMS Prigent et al. (2007) | 0.25° ~25km | Multi sensor: AVHRR, SSM/I, Scatterometer ERS | 2.1 – 5.9 | 1.4 – 4% |
| Fan et al. (2013) | 30 arcsec ~1km | Groundwater modelling | ~19.3[†] | ~17% |
| GLOWABO Verpoorter et al. (2014) | Shapefiles of lakes larger than 0.002 km² | Satellite imagery: Landsat and SRTM topography | 5 | 3.7% |
| SWAMPS Schroeder et al. (2015) | 25 km | Modeling using multi sensor info: SSM/I, SSM/S, QuikSCAT, ASCAT | 7.7 – 12.5[§] | 5.2 – 8.5% |
| ESA-CCI land cover Herold et al. (2015) | 10 arcsec ~300m | Multi sensor: SPOT vegetation, MERIS products | 6.1 | 4.7% |
| GIEMS-D15 Fluet-Chouinard et al. (2015) | 15 arcsec ~460m | Multi-sensor: SSM/I, ERS-1, AVHRR, Downscaled from a 0.25° wetland map | 6.5 – 17.3 | 5.0 - 13.2% |
| G3WBM Yamazaki et al. (2015) | 3 arcsec ~90m | Satellite imagery: Landsat | 3.2 | 2.5% |
| JRC Surface water Pekel et al. (2016) | 1 arcsec ~30m | Satellite imagery: Landsat, including maximum water extent and interannual occurrence | 2.8 – 4.4 | 2.1 - 3.4% |
| HydroLAKES Messager et al. (2016) | Shapefiles of lakes larger than 0.1 km² | Multiple inventory compilation including Canadian hydrographic dataset and SWBD | 2.7 | 1.8% |
| Hu et al. (2017) | 1 km | Development based on topographic wetness index and land-cover | 29.8[¶] | 22.5% |
| Poulter et al. (2017) | 0.5° ~50km | Merging SWAMPS and GLWD-3 | 10.5 | 7.1% |

**\* Percentages are those from the corresponding journal article or book. If no mention of percentage coverage exists, the value is calculated by dividing the wetland area by the land surface area excluding Antarctica, the glaciated Greenland and lakes.**

**† Excluding Caspian sea and large lakes**

**‡ Excluding Antarctica, glaciated Greenland, including lakes and Caspian sea. Additionally the range in GLWD is different based on interpretation of fractional wetlands.**

**§ Excluding large water bodies**

**¶ Including the Caspian sea**

**Table 2: Layers of wetlands constructed in the paper, their definition and the subsection where they are explained. Total land area for wetland percentages excludes lakes, Antarctica and the Greenland ice sheet.**

| Layer | | | Definition | Wetland percentage | Explained in |
|---|---|---|---|---|---|
| **RFW** (Regularly Flooded Wetlands) | | | Union of three inundation datasets (ESA-CCI, GIEMS-D15, JRC surface water) | 9.7% | Sect. 3.1 |
| **GDW** (Groundwater Driven Wetland) | **WTD** | | Pixels with water table depth less than 20 cm (Fan et al. 2013) | 15% | Sect. 3.2.1 |
| | **TI** | **(6%)** | Pixels with highest Tis, covering 15% of total land when combined with RFW | 6% | Sect. 3.2.2 |
| | | **(15%)** | Pixels with highest TIs values covering 15% of land | 15% | |
| | **TCI** | **(6.6%)** | Pixels with highest TCIs, covering 15% of total land when combined with RFW | 6.6% | |
| | | **(15%)** | Pixels with highest TCI values covering 15% of land | 15% | |
| | **TCTrI** | **(6%)** | Pixels with highest TCTrI, covering 15% of total land when combined with RFW | 6% | |
| | | **(15%)** | Pixels with highest TCTrI values covering 15% of land | 15% | |
| **CW** (Composite Wetland) | **WTD** | | Union of RFW and GDW-WTD | 21.1% | Sect. 3.3 |
| | **TI** | **(6%)** | Union of RFW and GDW-TI(6%) | 15% | |
| | | **(15%)** | Union of RFW and GDW-TI(15%) | 22.2% | |
| | **TCI** | **(6.6%)** | Union of RFW and GDW-TCI(6.6%) | 15% | |
| | | **(15%)** | Union of RFW and GDW-TCI(15%) | 21.6% | |
| | **TCTrI** | **(6%)** | Union of RFW and GDW-TCTrI(6%) | 15% | |
| | | **(15%)** | Union of RFW and GDW-TCTrI(15%) | 22.3% | |

**Table 3: Percent of overlap between GDW and RFW (percent of total pixels).**

| Groundwater-driven wetland layer | Intersecting with RFW | Non-intersecting with RFW |
|---|---|---|
| GDW-TI(6%) | 0.7% | 5.3% |
| GDW-TCI(6.6%) | 1.3% | 5.3% |
| GDW-TCTrI(6%) | 0.7% | 5.3% |
| GDW-TI(15%) | 2.5% | 12.5% |
| GDW-TCI(15%) | 3.6% | 11.4% |
| GDW-TCTrI(15%) | 2.4% | 12.6% |
| GDW-WTD(15%) | 3.8% | 11.2% |

**Table 4: Correlation between the developed and reference datasets (wetland fractions in 3 arc-min grid-cells). The highest three values in each column are shown in bold format, and grey cells give the values used in Fig. 4.**

| Dataset name | ESA-CCI | GIEMS-D15 | JRC surface water | RFW | GLWD-3 | GDW-WTD | Hu et al. (2017) |
|---|---|---|---|---|---|---|---|
| GDW-TI(15%) | -0.07 | 0.11 | 0.03 | 0.04 | 0.23 | 0.18 | 0.31 |
| GDW-TCTrI(15%) | -0.04 | -0.01 | -0.10 | 0.01 | 0.17 | 0.26 | 0.26 |
| GDW-TCI(15%) | 0.12 | 0.24 | 0.03 | 0.23 | 0.23 | 0.53 | 0.33 |
| GDW-WTD | 0.27 | 0.29 | 0.07 | 0.30 | **0.36** | **1.00** | **0.45** |
| CW-TI(6%) | 0.56 | 0.59 | 0.44 | **0.91** | 0.21 | 0.34 | 0.33 |
| CW-TCTrI(6%) | 0.49 | 0.59 | 0.43 | 0.78 | 0.24 | 0.43 | 0.40 |
| CW-TCI(6.6%) | 0.58 | 0.64 | 0.40 | **0.80** | 0.26 | 0.52 | 0.31 |
| CW-TI(15%) | 0.63 | 0.60 | 0.28 | 0.57 | 0.31 | 0.40 | 0.32 |
| CW-TCTrI(15%) | 0.55 | 0.45 | 0.36 | 0.51 | 0.32 | 0.38 | 0.28 |
| **CW-TCI(15%)** | **0.70** | **0.71** | **0.47** | 0.69 | 0.28 | **0.58** | 0.35 |
| **CW-WTD** | 0.63 | **0.69** | 0.37 | 0.65 | 0.34 | **0.65** | **0.43** |
| ESA-CCI | **1.00** | 0.33 | **0.66** | 0.53 | 0.28 | 0.27 | 0.27 |
| GIEMS-D15 | 0.33 | **1.00** | 0.36 | 0.67 | 0.26 | 0.29 | 0.20 |
| JRC surface water | **0.66** | 0.36 | **1.00** | 0.40 | 0.07 | 0.07 | 0.07 |
| RFW | 0.53 | 0.67 | 0.40 | **1.00** | **0.38** | 0.30 | 0.22 |
| GLWD-3 | 0.28 | 0.26 | 0.07 | 0.26 | **1.00** | 0.36 | 0.33 |
| Hu et al. (2017) | 0.27 | 0.20 | 0.07 | 0.22 | 0.33 | 0.45 | **1.00** |

[Figure]

**Fig. 1: Density of lakes regularly flooded wetlands and the three composing component (percent area in 3 arc-min grid-cells)**    **5**

[revised manuscript text omitted]